# ACQL: AN ADAPTIVE CONSERVATIVE Q-LEARNING FRAMEWORK FOR OFFLINE REINFORCEMENT LEARNING

## ABSTRACT

Offline Reinforcement Learning (RL), which relies only on static datasets without additional interactions with the environment, provides an appealing alternative to learning a safe and promising control policy. Most existing offline RL methods did not consider relative data quality and only crudely constrained the distribution gap between the learned policy and the behavior policy in general. Moreover, these algorithms cannot adaptively control the conservative level in more fine-grained ways, like for each state-action pair, leading to a performance drop especially over highly diversified datasets. In this paper, we propose an Adaptive Conservative Q-Learning (ACQL) framework that enables more flexible control over the conservative level of Q-function for offline RL. Specifically, we present two adaptive weight functions to shape the Q-values for collected and out-of-distribution data. Then we discuss different conditions under which the conservative level of the learned Q-function changes and define the monotonicity with respect to data quality and similarity. Motivated by the theoretical analysis, we propose a novel algorithm with the ACQL framework, using neural networks as the adaptive weight functions. To learn proper adaptive weight functions, we design surrogate losses incorporating the conditions for adjusting conservative levels and a contrastive loss to maintain the monotonicity of adaptive weight functions. We evaluate ACQL on the commonly-used D4RL benchmark and conduct extensive ablation studies to illustrate the effectiveness and state-of-the-art performance compared to existing offline DRL baselines.

## 1 INTRODUCTION

With the help of deep learning, Reinforcement Learning (RL) has achieved remarkable results on a variety of previously intractable problems, such as playing video games (Silver et al., 2016), controlling robot (Kalashnikov et al., 2018; Akkaya et al., 2019) and driving autonomous cars (Yu et al., 2020a; Zhao et al., 2022). However, the prerequisite that the agent has to interact with the environments makes the learning process costly and unsafe for many real-world scenarios. Recently, offline RL (Lange et al., 2012; Prudencio et al., 2022) has been proposed as a promising alternative to relax this requirement. In offline RL, the agent directly learns a control policy from a given static dataset, which is previously-collected by an unknown behavior policy. Offline RL enables the agent to achieve comparable or even better performance without additional interactions with environment.

Unfortunately, stripping the interactions from the online RL, offline RL is very challenging due to the distribution shift between the behavior policy and the learned policy. It often leads to the overestimation of values of out-of-distribution (OOD) actions (Kumar et al., 2019; Levine et al., 2020) and thus misleads the policy into choosing these erroneously estimated actions. To alleviate the distribution shift problem, recent methods (Kumar et al., 2019; Jaques et al., 2019; Wu et al., 2019; Siegel et al., 2020) proposed to constrain the learned policy to the behavior policy in different ways, such as limiting the action space (Fujimoto et al., 2019), using KL divergence (Wu et al., 2019) and using Maximum Mean Discrepancy (MMD) (Kumar et al., 2019). Besides directly constraining the policy, other methods (Kumar et al., 2020; Yu et al., 2021a;b; Ma et al., 2021) choose to learn a conservative Q-function to constrain the policy implicitly and thus alleviate the overestimation problem of Q-function.

However, most previous methods optimize all transition samples equally rather than selectively adapting, which may be overconservative especially for those non-expert datasets with high data diversity. As shown in Figure 1, a more conservative (with a larger alpha) CQL (Kumar et al., 2020) agent achieves higher returns on expert dataset while suffering from performance degradation on the random dataset, indicating for high-quality data, higher conservative level works better and vice versa. It clearly shows the importance of the conservative level on the final results. Therefore, it is more proper to use adaptive weights for different transition samples to control the conservative level, such as raising the Q-values more for good actions and less for bad actions.

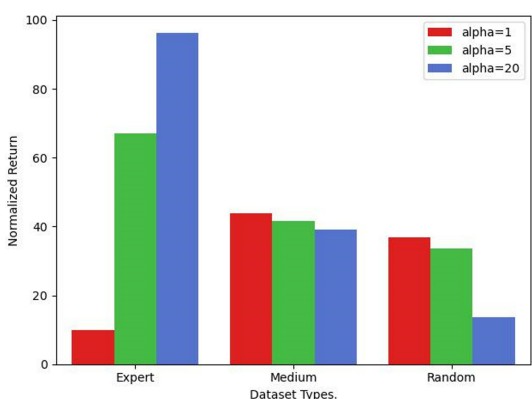

Figure 1: Performance gaps of CQL with different conservative levels on HalfCheetah tasks.

In this paper, we focus on how to constrain the Q-function in a more flexible way and propose a general Adaptive Conservative Q-Learning (ACQL) framework, which sheds the light on how to design a proper conservative Q-function. To achieve more fine-grained control over the conservative level, we use two adaptive weight functions to estimate the conservative weights for each transition sample. In the proposed framework, the form of the adaptive weight functions is not fixed, and we are able to define particular forms according to practical needs. We theoretically discuss in detail that the correlation between the different conservative levels and their corresponding conditions that the weight functions need to satisfy. We also formally define the monotonicity of the weight functions to depict the property that weight functions should raise the Q-values more for good actions and less for bad actions.

With the guidance of theoretical conditions, we propose one practical algorithm with learnable neural networks as adaptive weight functions. Overall, ACQL consists of three components. Firstly, we preprocess the fixed dataset to calculate the transition quality measurements, showing the data quality and similarity as pseudo labels. Then, with the help of the measurements, we construct surrogate losses to keep the conservative level of ACQL between the true Q-function and CQL. We also add contrastive loss to maintain the monotonicity of adaptive weight functions with respect to data quality and similarity. Lastly, we train the adaptive weight functions, actor network and critic network alternatively.

We summarize our contributions as follows: 1) We propose a more flexible framework ACQL that supports the fine-grained control of the conservative level in offline DRL. 2) We theoretically analyze how the conservative level changes conditioned on different forms of adaptive weight functions. 3) With the guidance of the proposed framework, we present a novel practical algorithm with carefully designed surrogate and contrastive losses to control the conservative levels and monotonicity. 4) We conduct extensive experiments on the D4RL benchmark and the state-of-the-art results well demonstrate the effectiveness of our framework.

## 2 RELATED WORK

**Imitation Learning.** To learn from a given static dataset, Imitation Learning (IL) is the most straightforward strategy. The core spirit of IL is to mimic the behavior policy. As the simplest form, behavior cloning still hold a place for offline reinforcement learning, especially for expert dataset. However, having expert datasets is only a minority of cases. Recently some methods (Chen et al., 2020; Siegel et al., 2020; Wang et al., 2020; Liu et al., 2021) aim to filter sub-optimal data and then apply the supervised learning paradigm afterward. Specifically, BAIL (Chen et al., 2020) performed imitation learning only on a high-quality subset of dataset purified by a learned value function. However, these methods often neglect the information contained in the bad actions with lower returns and thus often fail in tasks with non-optimal datasets. We believe these methods are

beneficial to our framework ACQL, since they split different data regions where different conservative levels can be set.

**Model-free Offline RL.** A large number of model-free offline RL methods aim to maximize the returns while constraining the learned policy and the behavior policy to be close enough. There are various ways for the direct constraint on policy including minimizing the KL-divergence (Jaques et al., 2019; Wu et al., 2019; Zhou et al., 2020), MMD (Kumar et al., 2019), or Wasserstein distance (Wu et al., 2019), and adding behavior cloning regularization (Fujimoto & Gu, 2021). The policy can be also constrained implicitly by actions space reduction (Fujimoto et al., 2019), importance sampling based algorithms (Sutton et al., 2016; Nachum et al., 2019), the implicit form of KL-divergence (Nair et al., 2020; Peng et al., 2019; Simão et al., 2020),uncertainty quantification (Agarwal et al., 2020; Kumar et al., 2019) or a conservative Q-function (Kumar et al., 2020; Ma et al., 2021; Sinha et al., 2022). More recently, Onestep RL (Brandfonbrener et al., 2021) and IQL (Kostrikov et al., 2021b) proposed to improve the policy after the convergence of Q functions. Trajectory Transformer (TT) (Janner et al., 2021) and Decision Transformer (DT) (Chen et al., 2021) leverages the advantage of Transformer (Vaswani et al., 2017) to optimize on trajectories. In this work, we propose a flexible framework ACQL for the model-free methods that constrain on Q-functions. ACQL supports defining different conservative levels for Q-function over each state-action pair, where CQL (Kumar et al., 2020) is one special case when all levels equal.

**Model-based Offline RL.** Recently, model-based methods (Janner et al., 2019; Kidambi et al., 2020; Yu et al., 2020b; 2021b; Matsushima et al., 2020) attract much attention for offline RL. They first learn the transition dynamics and reward function as a proxy environment which can be subsequently used for policy search. Given the proxy environment, offline methods (Ross & Bagnell, 2012; Kidambi et al., 2020), or run planning and trajectory optimization like LQR (Tassa et al., 2012) and MCTS (Browne et al., 2012) can be directly used for controlling. Although model-based offline RL can be highly sample efficient, direct use of it can be challenging due to distribution shift issue. In this paper, we only focus on model-free offline RL.

## 3 PROBLEM STATEMENT

We consider the environment as a fully-observed Markov Decision Process (MDP), which is represented by a tuple $(\boldsymbol{S}, \boldsymbol{A}, \boldsymbol{P}, r, \rho_0, \gamma)$. The MDP consists of the state space $\boldsymbol{S}$, the actions space $\boldsymbol{A}$, the transition probability distribution function $\boldsymbol{P} : \boldsymbol{S} \times \boldsymbol{A} \times \boldsymbol{S} \to [0, 1]$, the reward function $r : \boldsymbol{S} \times \boldsymbol{A} \times \boldsymbol{S} \to \mathbb{R}$, the initial state distribution $\rho_0(s)$ and the discount factor $\gamma \in (0, 1)$. The goal is to learn a control policy $\pi(a|s)$ that maximizes the cumulative discounted return $G_t = \sum_{t=0}^{\infty} \gamma^t r(s_t, a_t, s_{t+1}|s_0 \sim \rho_0, a_t \sim \pi(\cdot|s_t), s_{t+1} \sim \boldsymbol{P}(\cdot|s_t, a_t))$. In the Actor-Critic framework, the learning process repeatedly alternates between the policy evaluation that computes the value function for a policy and the policy improvement that obtains a better policy from the value function. Given a current replay buffer (dataset) $\boldsymbol{D} = \{(s, a, r, s')\}$ consisting of finite transition samples, the policy evaluation is defined as follows:

$$\hat{Q}^{k+1} \leftarrow \arg\min_{Q} \mathbb{E}_{s,a,s' \sim \boldsymbol{D}} \left[ \left( (\boldsymbol{B}^\pi \hat{Q}^k(s, a) - Q(s, a) \right)^2 \right], \quad (1)$$

where $k$ is the iteration number and the Bellman operator is defined as $\boldsymbol{B}^\pi \hat{Q}^k(s, a) = r(s, a) + \gamma \mathbb{E}_{a' \sim \pi^k(a'|s')}[\hat{Q}^k(s', a')]$. Note that the empirical Bellman operator $\hat{\mathcal{B}}^\pi$ is used in practical, which backs up only one transition, because it is difficult to contain all possible transitions $(s, a, s')$ in $\boldsymbol{D}$, especially for continuous action space. After approximating the Q-function, the policy improvement is performed as the following:

$$\hat{\pi}^{k+1} \leftarrow \arg\max_{\pi} \mathbb{E}_{s \sim \boldsymbol{D}, a \sim \pi^k(a|s)} \left[ \hat{Q}^{k+1}(s, a) \right]. \quad (2)$$

Compared to online RL, offline RL only allows learning from a fixed dataset $\boldsymbol{D}$ collected by an unknown behavior policy $\pi_\beta$, while prohibiting additional interactions with the environment. One of core issues in offline RL is the existence of the action distribution shift during training (Kumar et al., 2019; Wu et al., 2019; Jaques et al., 2019; Levine et al., 2020).

## 4 ADAPTIVE CONSERVATIVE Q-LEARNING FRAMEWORK (ACQL)

In this section, we propose a general framework, Adaptive Conservative Q-Learning (ACQL), which enables more flexible control over the conservative level of Q-function, compared to other Q-function constrained algorithms (Kumar et al., 2020; Ma et al., 2021; Yu et al., 2021a;b). Without loss of generality, we can consider the dataset collected by the behavior policy usually contains the data with both high and low returns, even though the behavior policy is a random policy. At the same time, among the actions sampled from the particular distribution $\mu$, there are also (relatively) good and bad actions instead of all actions having the same returns. In that case, we need a more flexible and fine-grained control method to constrain the Q-function for each state-action pair. Towards our goal, we propose to use two adaptive weight functions $d_\mu(s, a)$ and $d_{\pi_\beta}(s, a)$ to control the conservative level over the distribution $\mu$ and empirical behavior policy $\hat{\pi}_\beta$, respectively. Now the family of optimization problems of our framework ACQL is presented as below:

$$\min_Q \max_\mu \left( \mathbb{E}_{s \sim \boldsymbol{D}, a \sim \mu(a|s)} \left[ d_\mu(s, a) \cdot Q(s, a) \right] - \mathbb{E}_{s \sim \boldsymbol{D}, a \sim \hat{\pi}_\beta(a|s)} \left[ d_{\pi_\beta}(s, a) \cdot Q(s, a) \right] \right)$$

$$+ \frac{1}{2} \mathbb{E}_{s,a,s' \sim \boldsymbol{D}} \left[ \left( Q(s, a) - \hat{\mathcal{B}}^\pi \hat{Q}^k(s, a) \right)^2 \right] + \boldsymbol{R}(\mu). \quad (3)$$

Note that the form of the adaptive weight functions $d_\mu(s, a)$ and $d_{\pi_\beta}(s, a)$ is not fixed and can be customized according to different situations. It is their arbitrary form that supports us in shaping the Q-function more finely. In the following, we discuss the conditions about how we can adjust the conservative level of ACQL compared to the true Q-function and CQL (Kumar et al., 2020), and the properties that the adaptive weight functions $d_\mu(s, a)$ and $d_{\pi_\beta}(s, a)$ should have. First, we list different conditions on which ACQL is more conservative than the true Q-function in different levels in Proposition 4.1.

**Proposition 4.1** *(The conservative level of ACQL). For any $\mu$ with $\operatorname{supp} \mu \subset \operatorname{supp} \hat{\pi}_\beta$, without considering the sampling error between the empirical $\hat{\mathcal{B}}^\pi \hat{Q}$ and true Bellman backups $\boldsymbol{B}^\pi \hat{Q}$, the conservative level of ACQL can be controlled over the Q-values. The learned Q-function $\hat{Q}^\pi$ is more conservative than the true Q-function $Q^\pi$ point-wise, if:*

$$\forall s \in \boldsymbol{D}, a, \quad \frac{d_\mu \cdot \mu - d_{\pi_\beta} \cdot \pi_\beta}{\pi_\beta} \geq 0. \quad (4)$$

Moreover, as shown in Appendix A, we can also control the conservative level over other regions like the V-values or empirical MDP by changing the state, action space in Equation (4). As a special instance of our framework ACQL, CQL (Kumar et al., 2020) proposes to constraint the conservative level over the excepted V-values. And CQL performs the optimization for all state-action pairs with the same weight $\alpha$, which may be too rigid and over conservative for some scenarios. Now, if we replace the "$\geq$" to "$\leq$" in Equation (4), we can get the conditions on which ACQL is less conservative than the true Q-function. Following (Auer et al., 2008; Osband et al., 2016; Kumar et al., 2020), we also show that ACQL bounds the gap between the learned Q-values and true Q-values with the consideration of the sampling error between the empirical $\hat{\mathcal{B}}^\pi \hat{Q}$ and actual Bellman operator $\boldsymbol{B}^\pi \hat{Q}$. Due to the space limitation, we present the proposition and proof in Appendix A. Besides the comparison to the true Q-function, we also give a theoretical discussion about the comparison to CQL (Kumar et al., 2020) in the following.

**Proposition 4.2** *(The conservative level compared to CQL). For any $\mu$ with $\operatorname{supp} \mu \subset \operatorname{supp} \hat{\pi}_\beta$, given the Q-function learned from CQL is $\hat{Q}^\pi_{CQL}(s, a) = Q^\pi - \alpha \frac{\mu - \pi_\beta}{\pi_\beta}$, similiar as in 4.1, the conservative level of ACQL compared to CQL can be controlled over the Q-values. The learned Q-function $\hat{Q}^\pi$ is less conservative than the CQL Q-function $\hat{Q}^\pi_{CQL}$ point-wise, if:*

$$\forall s \in \boldsymbol{D}, a, \quad \frac{(\alpha - d_\mu)\mu - (\alpha - d_{\pi_\beta})\pi_\beta}{\pi_\beta} \geq 0. \quad (5)$$

Besides the discussion about the conservative level of ACQL, to depict the property that a good action should have a higher Q-value than a bad action, we formally define the monotonicity of the adaptive weight functions as following:

**Definition 4.1** *(The monotonicity of the adaptive weight functions). For any state $s \in \boldsymbol{D}$, the monotonicity of the adaptive weight functions is defined as that a good action $a \in \boldsymbol{A}$ with a higher true Q value has a lower $d_\mu$ value and higher $d_{\pi_\beta}$ value:*

$$\forall s_i, s_j \in \boldsymbol{D}, a_i, a_j \in \mu(a|s), d_\mu(s_i, a_i) - d_\mu(s_j, a_j) \propto Q^*(s_j, a_j) - Q^*(s_i, a_i), \quad (6)$$

$$\forall s_i, s_j \in \boldsymbol{D}, a_i, a_j \in \hat{\pi}_\beta(a|s), d_{\pi_\beta}(s_i, a_i) - d_{\pi_\beta}(s_j, a_j) \propto Q^*(s_i, a_i) - Q^*(s_j, a_j), \quad (7)$$

where $Q^*$ is the optimal Q-function. In Definition 4.1, we use the optimal Q-function which is a natural ideal metric to measure the action quality to define the monotonicity. However, it is a ill-posed problem since if we know the optimal Q-function, we can directly train an optimal policy and solve the problem. And we also do not know what the optimal proportional relationship should be. In the next section 5, we construct a contrastive loss using transition quality measurements to approximate this property for ACQL. All the proofs are provided in Appendix A.

## 5 ACQL WITH LEARNABLE WEIGHT FUNCTIONS

In this section, derived from the theoretical discussion, we propose one practical ACQL algorithm in the guidance of theoretical conditions with learnable neural network as adaptive weight functions. To adaptively control the conservative level, there are three steps for ACQL. Firstly, we preprocess the fixed dataset to calculate the transition quality measurements. Then, with the help of the transition quality measurements, we construct surrogate losses to control the conservative level of ACQL. We also add contrastive losses to maintain the monotonicity. Lastly, we train the adaptive weight functions, actor network and critic network alternatively.

### 5.1 TRANSITION QUALITY MEASUREMENTS

To seek a replacement for the optimal Q-function $Q^*$ in Definition 4.1, we firstly preprocess the fixed dataset and calculate the transition quality measurements for each state-action pair. Note that different tasks may have different magnitudes of Q-values, thus it is better to seek a normalized measurement ranging in $(0, 1)$. For the action in the fixed dataset, we define Relative Transition Quality $m(s, a)$ by combining both the single step reward and the whole discounted trajectory return to measure the data quality:

$$\forall (s, a) \in \boldsymbol{D}, m(s, a) = \frac{1}{2}(r_{norm}(s, a) + g_{norm}(s, a)), \quad (8)$$

where $r_{norm} = \frac{r_{cur} - r_{min}}{r_{max} - r_{min}}$ and $g_{norm} = \frac{g_{cur} - g_{min}}{g_{max} - g_{min}}$. The $r_{min}, r_{max}, g_{min}, g_{max}$ are the minimum and maximum values of the single step reward and the Monte Carlo return of the whole trajectory in the dataset respectively. Note that the range of $m(s, a)$ is $(0, 1)$ and the higher $m(s, a)$ indicates that $a$ is a better action. For the OOD actions, since they do not appear in the dataset and we are not able to use the single rewards and Monte Carlo returns, we choose to use both the quality of its corresponding in-dataset action $m(s, a_{in})$ and the Euclidean distance between the OOD action and action in the dataset with the same state.

$$\forall (s, a_{in}) \in \boldsymbol{D}, a_\mu \in \mu(a|s), m(s, a_\mu) = \frac{1}{2}\left(m(s, a_{in}) - \frac{1}{2}||a_\mu - a_{in}||_2 + 1\right). \quad (9)$$

Consistent with $m(s, a_{in})$, we shift and scale in Equation 9 to make the range of $m(s, a_\mu)$ is also $(0, 1)$ and the higher $m(s, a_\mu)$ indicates that $a_\mu$ is a better action. Note that $m(s, a_\mu)$ is calculated over every training batch. We argue that the above Equations 8 and 9 is only a simple and effective way to be the replacement of the optimal Q-function and it can be served as a baseline for future algorithms. Many methods including the "upper envelope" in BAIL (Chen et al., 2020) and uncertainty estimation in (Yu et al., 2020b) have the potential to be incorporated in ACQL.

### 5.2 OPTIMIZATION TO CONTROL THE CONSERVATIVE LEVEL

Suppose we present the functions $d_\mu, d_{\pi_\beta}$ as deep neural networks, the key point is how to design the loss functions. Recapping the Equations (4) and (5), we have known the conditions for different conservative level and thus adapt them to the loss functions. More specifically, we incorporate

the conditions, which aims to learn a Q-funtion more conservative than true Q-function but less conservative than CQL, into the hinge losses as shown in the following:

$$\boldsymbol{L}_{cl\_true}(d_\mu, d_{\pi_\beta}) = max(0, d_{\pi_\beta} \cdot \pi_\beta - d_\mu \cdot \mu + C_1 \cdot \pi_\beta), \tag{10}$$

$$\boldsymbol{L}_{cl\_cql}(d_\mu, d_{\pi_\beta}) = max(0, (d_\mu - \alpha) \cdot \mu - (d_{\pi_\beta} - \alpha) \cdot \pi_\beta + C_2 \cdot \pi_\beta), \tag{11}$$

where $C_1$ and $C_2$ are used to control the soft margin of the conservative level compared to the true Q-function and CQL respectively. A higher $C_1$ means a more conservative Q-function than true Q-function, while a higher $C_2$ represents a less conservative Q-function than CQL. However, $C_1$ and $C_2$ are difficult to tune and not adaptive enough as fixed hyperparameters. We leverage the relative transition quality to calculate $C_1$ and $C_2$ automatically based on the property in Definition 4.1 that a good action should have less conservative Q-value and thus a lower $C_1$ and a higher $C_2$.

$$C_1(s, a) = (1 - m(s, a)) \cdot r_{max}, \tag{12}$$

$$C_2(s, a) = m(s, a) \cdot r_{max}. \tag{13}$$

The ranges of $C_1$ and $C_2$ are both $(0, r_{max})$. Note that $C_1$ and $C_2$ are the soft margin over the Q-values, thus we use the maximum single reward $r_{max}$ as the maximum margin to avoid excessive fluctuations in policy evaluation. Nevertheless, during the optimization for $\boldsymbol{L}_{cl\_true}$ and $\boldsymbol{L}_{cl\_cql}$, we find it is prone to cause arithmetic underflow since the $log_\mu, log_{\pi_\beta}$ is usually very small like -1000, then the resulting $\mu(a|s)$ and $\pi_\beta(a|s)$ become 0 after exponentiation operation. To avoid the arithmetic underflow problem, we use a necessity of Equations (4) and (5) to form surrogate losses.

**Lemma 5.1** *For $x > 0$, $\ln x \le x - 1$. $\ln x = x - 1$ if and only if $x = 1$.*

The proof is provided in Appendix A. Then the resulting surrogate losses are the following:

$$\boldsymbol{L}_{cl\_true}(d_\mu, d_{\pi_\beta}) = max(0, d_{\pi_\beta} \cdot (\ln \pi_\beta + 1) - d_\mu \cdot (\ln \mu + 1) + C_1 \cdot (\ln \pi_\beta + 1)), \tag{14}$$

$$\boldsymbol{L}_{cl\_cql}(d_\mu, d_{\pi_\beta}) = max(0, (d_\mu - \alpha)(\ln \mu + 1) - (d_{\pi_\beta} - \alpha)(\ln \pi_\beta + 1) + C_2(\ln \pi_\beta + 1)). \tag{15}$$

### 5.3 OPTIMIZATION TO MAINTAIN THE MONOTONICITY

Besides the surrogate losses to control the conservative levels of ACQL, we also construct contrastive loss to main the monotonicity as stated in Definition 4.1. We use Mean Squared Error (MSE) for simplicity. The contrastive loss is defined as the following:

$$\forall (s_i, a_i), (s_j, a_j) \notin \boldsymbol{D}, (s_k, a_k), (s_l, a_l) \in \boldsymbol{D}, \tag{16}$$

$$\boldsymbol{L}_{mono}(d_\mu, d_{\pi_\beta}) = \|(\sigma(d_\mu(s_i, a_i)) - \sigma(d_\mu(s_j, a_j))) - (\sigma(m(s_j, a_j)) - \sigma(m(s_i, a_i)))\|_2^2 \tag{17}$$

$$+ \|(\sigma(d_{\pi_\beta}(s_k, a_k)) - \sigma(d_{\pi_\beta}(s_l, a_l))) - (\sigma(m(s_k, a_k)) - \sigma(m(s_l, a_l)))\|_2^2, \tag{18}$$

where we use the softmax operation $\sigma(\cdot)$ over the current batch of the training data to unify the orders of magnitude between the adaptive weights and transition quality measurements.

### 5.4 FINAL OBJECT

To learn a conservative Q-function, the adaptive weight should be positive, and thus we add a regularizer term as the following:

$$\boldsymbol{L}_{pos}(d_\mu, d_{\pi_\beta}) = max(0, -d_\mu) + max(0, -d_{\pi_\beta}), \tag{19}$$

The final object for training the adaptive weight functions $d_\mu, d_{\pi_\beta}$ is:

$$\boldsymbol{L}(d_\mu, d_{\pi_\beta}) = \boldsymbol{L}_{cl\_true}(d_\mu, d_{\pi_\beta}) + \boldsymbol{L}_{cl\_cql}(d_\mu, d_{\pi_\beta}) + \boldsymbol{L}_{mono}(d_\mu, d_{\pi_\beta}) + \boldsymbol{L}_{pos}(d_\mu, d_{\pi_\beta}). \tag{20}$$

ACQL is built on the top of the CQL (Kumar et al., 2020), which sets $\mu = \pi$ and $\boldsymbol{R}(\mu) = -D_{KL}(\mu, Unif(a))$ in Equation 3 in Section 5. And the optimization problem becomes:

$$\min_Q \max_\pi \min_{d_\mu, d_{\pi_\beta}} \left( \mathbb{E}_{s \sim \boldsymbol{D}, a \sim \pi(a|s)} [d_\mu(s, a) \cdot Q(s, a)] - \mathbb{E}_{s \sim \boldsymbol{D}, a \sim \hat{\pi}_\beta(a|s)} [d_{\pi_\beta}(s, a) \cdot Q(s, a)] \right)$$

$$+ \frac{1}{2} \mathbb{E}_{s, a, s' \sim \boldsymbol{D}} \left[ \left( Q(s, a) - \hat{\mathcal{B}}^\pi \hat{Q}^k(s, a) \right)^2 \right] - \boldsymbol{D}_{KL}(\pi, Unif(a)) + \boldsymbol{L}(d_\mu, d_{\pi_\beta}). \tag{21}$$

During the training process, we add weight neural networks $d_\mu, d_{\pi_\beta}$ and train the weight networks, Q networks and policy networks alternatively. Due to the space limitation, the implementation details are provided in Appendix B.

Table 1: Normalized results on D4RL Gym-MuJoCo environments.

| Task Name | BC | BCQ | BEAR | BRAC-v | BRAC-p | AWR | TD3+BC | F-BRC | CQL | ACQL |
|---|---|---|---|---|---|---|---|---|---|---|
| halfcheetah-e-v0 | 107.0 | - | 108.2 | -1.1 | 3.8 | - | 105.7 | **108.4** | 103.5 | 97.7 |
| halfcheetah-m-e-v0 | 35.8 | 64.7 | 53.4 | 41.9 | 44.2 | 52.7 | **97.9** | 93.3 | 62.4 | 93.2 |
| halfcheetah-m-r-v0 | 38.4 | 38.2 | 38.6 | **47.7** | 45.4 | 40.3 | 43.3 | 43.2 | 46.2 | 46.4 |
| halfcheetah-m-v0 | 36.1 | 40.7 | 41.7 | **46.3** | 43.8 | 37.4 | 42.8 | 41.3 | 44.4 | 42.8 |
| halfcheetah-r-v0 | 2.1 | 2.2 | 25.1 | 31.2 | 24.1 | 2.5 | 10.2 | 33.3 | **35.4** | 31.3 |
| halfcheetah-sum | 219.4 | - | 267.0 | 166.0 | 161.3 | - | 299.9 | **319.5** | 291.9 | 311.4 |
| hopper-e-v0 | 109.0 | - | 110.3 | 3.7 | 6.6 | - | 112.2 | **112.3** | 112.2 | **116.3** |
| hopper-m-e-v0 | 111.9 | 110.9 | 96.3 | 0.8 | 1.9 | 27.1 | 112.2 | **112.4** | 98.7 | 111.8 |
| hopper-m-r-v0 | 11.8 | 33.1 | 33.7 | 0.6 | 0.6 | 28.4 | 31.4 | 35.6 | **48.6** | **77.3** |
| hopper-m-v0 | 29.0 | 54.5 | 52.1 | 31.1 | 32.7 | 35.9 | **99.5** | 99.4 | 58.0 | 85.8 |
| hopper-r-v0 | 9.8 | 10.6 | 11.4 | **12.2** | 11.0 | 10.2 | 11.0 | 11.3 | 10.8 | **12.7** |
| hopper-sum | 271.5 | - | 303.8 | 48.4 | 52.8 | - | 366.3 | **371.0** | 328.3 | **403.9** |
| walker-e-v0 | **125.7** | - | 106.1 | 0.0 | -0.2 | - | 105.7 | 103.0 | 107.2 | 116.1 |
| walker-m-e-v0 | 6.4 | 57.5 | 40.1 | 81.6 | 76.9 | 53.8 | 101.1 | 105.2 | **111.0** | 94.5 |
| walker-m-r-v0 | 11.3 | 15.0 | 19.2 | 0.9 | -0.3 | 15.5 | 25.2 | **41.8** | 26.7 | 45.2 |
| walker-m-v0 | 6.6 | 53.1 | 59.1 | **81.1** | 77.5 | 17.4 | 79.7 | 78.8 | 79.2 | **81.8** |
| walker-r-v0 | 1.6 | 4.9 | **7.3** | 1.9 | -0.2 | 1.5 | 1.4 | 1.5 | 7.0 | **11.8** |
| walker-sum | 151.6 | - | 231.8 | 165.5 | 153.7 | - | 313.1 | 330.3 | **331.1** | **349.4** |

# 6 EXPERIMENTS

## 6.1 COMPARISONS TO OFFLINE RL BASELINES

We conducted all the experiments on the commonly-used offline RL benchmark D4RL (Fu et al., 2020), which includes many task domains (Todorov et al., 2012; Brockman et al., 2016; Rajeswaran et al., 2017) and a variety of dataset types. Aiming to provide a comprehensive comparison, we compared ACQL to many state-of-the-art model-free algorithms including behavioral cloning (BC), SAC-off (Haarnoja et al., 2018), BEAR (Kumar et al., 2019), BRAC (Wu et al., 2019), AWR (Peng et al., 2019), BCQ (Fujimoto et al., 2019), aDICE (Nachum et al., 2019) TD3+BC (Fujimoto & Gu, 2021), Fisher-BRC (Kostrikov et al., 2021a), and CQL (Kumar et al., 2020). For the sake of fair comparisons, we directly reported the results of all baselines from the D4RL whitepaper (Fu et al., 2020) and their original papers. To be consistent with previous works, we trained ACQL for 1.0 M gradient steps and evaluated for 10 episodes every 1000 training iterations. The results are the average values of 10 episodes over 3 random seeds and are obtained from the workflow proposed by (Kumar et al., 2021).

**Gym-MuJoCo Tasks.** The Gym-MuJoCo tasks include "halfcheetah", "hopper" and "walker" with 5 kinds of dataset types of each, ranging from expert data to random data. For brevity, we marked "-expert", "-medium-expert", "-medium-replay", "-medium" and "random" as "-e", "-m-e", "-m-r", "-m" and "-r" respectively. Table 1 shows the normalized returns of all 15 Gym-MuJoCo version-0 tasks. We can observe that ACQL outperforms other baselines on Hopper and Walker environments by a large margin. From the perspective of the dataset types, ACQL is very a balanced algorithm that achieves excelling results on all kinds of datasets with expert, medium and random data. We also built additaional comparisons to more state-of-the-art model-free algorithms including Decision Transformer (DT) (Chen et al., 2021), AWAC (Nair et al., 2020), Onestep RL (Brandfonbrener et al., 2021), TD3+BC (Fujimoto & Gu, 2021), IQL (Kostrikov et al., 2021a) and CQL (Kumar et al., 2020) on version-2 datasets in Appendix C.1.

**Adroit Tasks.** The adroit tasks (Rajeswaran et al., 2017) are high-dimensional robotic manipulation tasks with sparse reward and include expert data and human demonstrations from narrow distributions. Table 2 shows the normalized returns of all 12 Kitchen tasks. ACQL delivers higher performance than other baselines on "cloned" datasets, i.e., with mixed expert data and human demonstrations.

**Franka Kitchen Tasks.** The Franka Kitchen tasks (Gupta et al., 2019) include complex trajectories as the offline datasets and aim to evaluate the "stitching" ability of the agent on a realistic kitchen environment. Table 3 shows the normalized returns of all 3 Kitchen tasks. We can see that ACQL consistently exceeds CQL, BC and other baselines on all 3 kinds of datasets, since ACQL can control the conservative level adaptively.

**AntMaze Tasks.** The AntMaze tasks mimic real-world robotic navigation tasks that aim to control an "Ant" quadruped robot to reach the goal location from a start location with only 0-1 sparse rewards given. Due to the space limitation, we reported the experimental results in Appendix C.3

Table 2: Normalized results on D4RL Adroit environments.

| Task Name | BC | SAC-Off | BEAR | BRAC-p | BRAC-v | AWR | BCQ | aDICE | CQL | ACQL |
|---|---|---|---|---|---|---|---|---|---|---|
| door-cloned-v0 | -0.1 | 0.0 | -0.1 | -0.1 | -0.1 | 0.0 | 0.0 | 0.0 | **0.4** | **6.6** |
| door-expert-v0 | 34.9 | 7.5 | **103.4** | -0.3 | -0.3 | 102.9 | 99.0 | 0.0 | 101.5 | 101.5 |
| door-human-v0 | 0.5 | 3.9 | -0.3 | -0.3 | -0.3 | 0.4 | 0.0 | 0.0 | **9.9** | 0.2 |
| hammer-cloned-v0 | 0.8 | 0.2 | 0.3 | 0.3 | 0.3 | 0.4 | 0.4 | 0.3 | **2.1** | **12.7** |
| hammer-expert-v0 | 125.6 | 25.2 | **127.3** | 0.3 | 0.3 | 39.0 | 107.2 | 0.3 | 86.7 | 77.3 |
| hammer-human-v0 | 1.5 | 0.5 | 0.3 | 0.3 | 0.2 | 1.2 | 0.5 | 0.3 | **4.4** | **10.6** |
| pen-cloned-v0 | **56.9** | 23.5 | 26.5 | 1.6 | -2.5 | 28.0 | 44.0 | -2.9 | 39.2 | 47.6 |
| pen-expert-v0 | 85.1 | 6.1 | 105.9 | -3.5 | -3.0 | 111.0 | **114.9** | -3.5 | 107.0 | **143.1** |
| pen-human-v0 | 34.4 | 6.3 | -1.0 | 8.1 | 0.6 | 12.3 | **68.9** | -3.3 | 37.5 | 54.7 |
| relocate-cloned-v0 | -0.1 | -0.2 | -0.3 | -0.3 | -0.3 | -0.2 | -0.3 | -0.3 | **-0.1** | **0.1** |
| relocate-expert-v0 | **101.3** | -0.3 | 98.6 | -0.3 | -0.4 | 91.5 | 41.6 | -0.1 | 95.0 | 60.4 |
| relocate-human-v0 | 0.0 | 0.0 | -0.3 | -0.3 | -0.3 | 0.0 | -0.1 | -0.1 | **0.2** | 0.1 |

Table 3: Normalized results on D4RL Franka Kitchen environments.

| Task Name | BC | SAC-Off | BEAR | BRAC-p | BRAC-v | AWR | BCQ | aDICE | CQL | ACQL |
|---|---|---|---|---|---|---|---|---|---|---|
| kitchen-complete-v0 | 33.8 | 15.0 | 0.0 | 0.0 | 0.0 | 0.0 | 8.1 | 0.0 | 43.8 | **50.2** |
| kitchen-mixed-v0 | 47.5 | 2.5 | 47.2 | 0.0 | 0.0 | 10.6 | 8.1 | 2.5 | 51.0 | **55.9** |
| kitchen-partial-v0 | 33.8 | 0.0 | 13.1 | 0.0 | 0.0 | 15.4 | 18.9 | 0.0 | 49.8 | **51.7** |

## 6.2 COMPARISONS AMONG DIFFERENT CONSERVATIVE LEVELS

To demonstrate more intuitively that ACQL can adaptively control the conservative level, we compared ACQL to CQL with different $\alpha$ values ranging from $1, 2, 5, 10, 20$. The different $\alpha$ values represent different conservative levels and a higher $\alpha$ means more conservative. Figure 2 plotted the learning curves of ACQL and CQL with 5 kinds of $\alpha$ on the Hopper-v0 environments. We also provided a more detailed Figure 4 and a Table 8 of the quantitative results in Appendix C.2. From the Figure 2, one trend we can clearly observe is that CQL with a higher conservative level can usually achieve higher performance on expert datasets as opposed to that a lower conservative level is more effective for random datasets. Moreover, it is difficult for CQL to achieve satisfactory performance on all kinds of dataset with a fixed $\alpha$. It is exactly the issue that the ACQL focuses on. Since ACQL generates adaptive weight for each state-action pair and control the conservative level in a more fine-grained way, ACQL can achieve balanced and state-of-the-art results for different dataset types.

Table 4 shows the comparisons of average Q-values over the datasets on HalfCheetah-v0 environments. We can observe that as the $\alpha$ of CQL increase, the learned Q-values decreases, since a higher $\alpha$ represents a higher conservative level. Compared to CQL, the Q-values learned by ACQL is higher than CQL showing ACQL is less conservative than CQL while the Q-values do not explode.

## 6.3 ABLATION STUDY

**Effect of the Proposed Losses.** Due to the space limitation, we reported the quantitative results on the Hopper environment including 5 kinds of datasets in Table 5. More results about other environments are provided in Appendix C.4. As shown in Table 5, when we only control the conservative level using $L_{cl\_true}$ (second row) or $L_{cl\_cql}$ (third row), ACQL dropped its performance seriously, with only around 21.0 of normalized returns even on expert dataset. Without using $L_{mono}$ and $L_{pos}$ to limit the range of the adaptive weights, it is prone to learn erroneously adaptive weights and the errors increase like a snowball as the policy evaluation repeats and lead to failure.

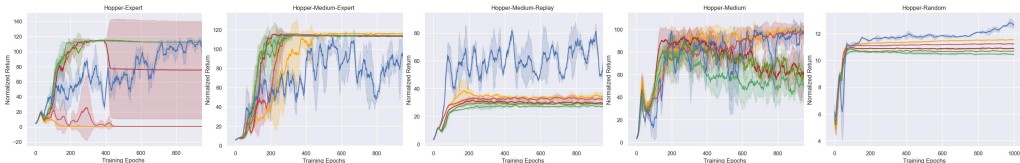

Figure 2: Learning curves comparing to CQL with different conservative levels on Hopper-v0 environments.

Table 4: Comparisons of average Q-values over the datasets on HalfCheetah-v0 environments.

| Task Name | CQL-1 | CQL-2 | CQL-5 | CQL-10 | CQL-20 | ACQL |
|---|---|---|---|---|---|---|
| halfcheetah-expert-v0 | 1276.3 | 1252.9 | 1221.9 | 1202.7 | 1178.3 | 1271.1 |
| halfcheetah-medium-expert-v0 | 901.9 | 803.6 | 627.5 | 545.7 | 439.1 | 755.1 |
| halfcheetah-medium-replay-v0 | 283.5 | 237.6 | 72.3 | -178.1 | -248.3 | 452.4 |
| halfcheetah-medium-v0 | 103.8 | -9.1 | -49.7 | -241.1 | -313.6 | 385.9 |
| halfcheetah-random-v0 | 154.6 | 126.7 | 76.9 | 40.5 | 29.4 | 304.1 |

Table 5: Ablation Study on Gym-MuJoCo Hopper-v0 environments in terms of normalized results.

| | $\mathcal{L}_{cl\_true}$ | $\mathcal{L}_{cl\_cql}$ | $\mathcal{L}_{mono}$ | $\mathcal{L}_{pos}$ | hopper-e | hopper-m-e | hopper-m-r | hopper-m | hopper-r |
|---|---|---|---|---|---|---|---|---|---|
| 1 | ✓ | | | | 21.7 | 8.3 | 33.1 | 52.8 | 10.6 |
| 2 | | ✓ | | | 21.5 | 31.5 | 38.2 | 58.4 | **24.3** |
| 3 | ✓ | ✓ | | | 34.8 | 85.4 | 66.9 | 41.7 | 12.5 |
| 4 | ✓ | ✓ | ✓ | | 114.4 | **112.4** | 71.5 | 56.3 | 10.7 |
| 5 | ✓ | ✓ | ✓ | ✓ | **116.3** | 111.8 | **77.3** | **85.8** | 12.7 |

**Visualization of the Adaptive Weights.** To further check the monotonicity of the adaptive weight functions, we visualized the adaptive weights and the corresponding relative transition quality measurements for HalfCheetah-random-v0 dataset in Figure 3. Note that all values are normalized to uniform the magnitudes and are sorted according to the quality measurements. In the left of Figure 3, $d_{\mu}\_rand$ are the $d_{\mu}(s, a)$ where $s$ are in dataset and $a$ are sampled randomly, while $d_{\mu}\_policy$ are the $d_{\mu}(s, a)$ where $a$ are predicted from the training policy. We can see the trend that as the quality measurements (green and red points) increase, the adaptive weights $d_{\mu}$ (blue and yellow points) decrease, indicating the Q-values of good OOD state-action pairs should be suppressed less. In the right of Figure 3, $d_{\pi_{\beta}}\_dataset$ are the $d_{\pi_{\beta}}(s, a)$ where $s, a$ are sampled from the dataset. As the the quality measurements (green points) increase, the adaptive weights $d_{\pi_{\beta}}$ (blue points) also increase, indicating the Q-values of good in-dataset state-action pairs should be higher.

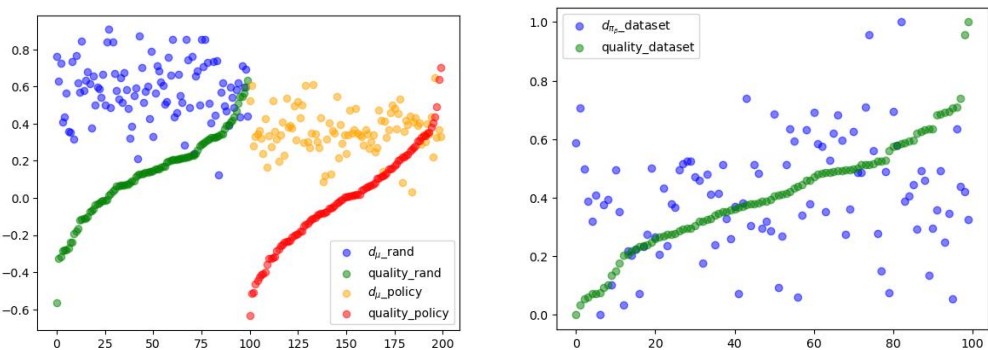

Figure 3: Visualization of the adaptive weights and relative transition quality $m(s, a)$ on HalfCheetah-random-v0 dataset.

## 7 CONCLUSION

In this paper, we proposed a flexible framework named Adaptive Conservative Q-Learning (ACQL), which sheds the light on how to control the conservative level of the Q-function in a fine-grained way. In ACQL, two weight functions, corresponding to the out-of-distribution (OOD) actions and actions in the dataset, are introduced to adaptively shape the Q-function. More importantly, the form of these two adaptive weight functions is not fixed and it is possible to define particular forms for different scenarios, e.g., elaborately hand-designed rules or learnable deep neural networks. We provide a detailed theory analysis about how the conservative level of the learned Q-function changes under different conditions and define the monotonicity of the adaptive weight functions. To illustrate the feasibility of our framework, we propose a novel practical algorithm using neural networks as the weight functions. With the guidance of the theoretical analysis, we construct two surrogate and contrastive losses to control the conservative level and maintain the monotonicity. We build extensive experiments on commonly-used offline RL benchmarks and the state-of-the-art results well demonstrate the effectiveness of our method.

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

## A  PROOFS

**Proposition A.1** *(The conservative level of ACQL). For any $\mu$ with $\operatorname{supp} \mu \subset \operatorname{supp} \hat{\pi}_\beta$, without considering the sampling error between the empirical $\hat{\mathcal{B}}^\pi \hat{Q}$ and true Bellman backups $\boldsymbol{B}^\pi \hat{Q}$, the conservative level of ACQL can be controlled at three levels according to different conditions:*
*1) Control over the Q-values. The learned Q-function $\hat{Q}^\pi_{ACQL}$ is more conservative than the true Q-function $Q^\pi$ point-wise, if:*

$$\forall s \in \boldsymbol{D}, a, \frac{d_\mu \cdot \mu - d_{\pi_\beta} \cdot \pi_\beta}{\pi_\beta} \geq 0. \tag{22}$$

*2) Control over the V-values. The excepted values of learned Q-function $\hat{Q}^\pi_{ACQL}$ is more conservative than excepted values of the true Q-function $Q^\pi$, if:*

$$\forall s \in \boldsymbol{D}, \sum_a \frac{d_\mu \cdot \mu - d_{\pi_\beta} \cdot \pi_\beta}{\pi_\beta} \geq 0. \tag{23}$$

*3) Control over the empirical MDP. The learned Q-function $\hat{Q}^\pi_{ACQL}$ is more conservative over the empirical MDP, if:*

$$\sum_{s \in \boldsymbol{D}} \sum_a \frac{d_\mu \cdot \mu - d_{\pi_\beta} \cdot \pi_\beta}{\pi_\beta} \geq 0. \tag{24}$$

**Proof of Proposition A.1**. Without considering the sampling error between the empirical $\hat{\mathcal{B}}^\pi \hat{Q}$ and true Bellman backups $\boldsymbol{B}^\pi \hat{Q}$, we first show the optimization problem of Q-function in ACQL as the following:

$$\hat{Q}^{k+1}_{ACQL} \leftarrow \min_Q \left( \mathbb{E}_{s \sim \boldsymbol{D}, a \sim \mu(a|s)} \left[ d_\mu(s, a) \cdot Q(s, a) \right] - \mathbb{E}_{s \sim \boldsymbol{D}, a \sim \pi_\beta(a|s)} \left[ d_{\pi_\beta}(s, a) \cdot Q(s, a) \right] \right)$$
$$+ \frac{1}{2} \mathbb{E}_{s, a, s' \sim \boldsymbol{D}} \left[ \left( Q(s, a) - \boldsymbol{B}^\pi \hat{Q}^k(s, a) \right)^2 \right] \tag{25}$$

By setting the derivative of Equation 25 to 0, we can obtain the form of the resulting Q-function $\hat{Q}^{k+1}$ in ACQL:

$$\forall s \in \boldsymbol{D}, a, \frac{\partial \hat{Q}^{k+1}_{ACQL}}{\partial Q} = 0 \tag{26}$$

$$\Rightarrow d_\mu \cdot \mu - d_{\pi_\beta} \cdot \pi_\beta + \pi_\beta \cdot (Q - \boldsymbol{B}^\pi \hat{Q}^k) = 0 \tag{27}$$

$$\Rightarrow \hat{Q}^{k+1}_{ACQL} = \boldsymbol{B}^\pi \hat{Q}^k - \frac{d_\mu \cdot \mu - d_{\pi_\beta} \cdot \pi_\beta}{\pi_\beta}. \tag{28}$$

Note that the true Q-function is only derived from the true Bellman operator $Q^{k+1} = \boldsymbol{B}^\pi \hat{Q}^k$. Based on the Equation 28, we have exactly the condition to control the conservative level over the Q-function as shown in Equation 22. If we want to relax the conservative level like requiring to control over the V-values or the empirical MDP, we can easily relax the Equation 28 to the integration over each state or the whole empirical MDP as shown in Equations (23) and (24) respectively. For the conditions where we want to make the learned Q-function $\hat{Q}^\pi_{ACQL}$ is less conservative than the true Q-function $Q^\pi$, we can easily replace the "$\geq$" to "$\leq$" in Proposition A.1.

Next we show that ACQL bounds the gap between the learned Q-values and true Q-values with the consideration of the sampling error between the empirical $\hat{\mathcal{B}}^\pi \hat{Q}$ and actual Bellman operator $\boldsymbol{B}^\pi \hat{Q}$. Following (Auer et al., 2008; Osband et al., 2016; Kumar et al., 2020), the error can be bounded by leveraging the concentration properties of $\hat{\mathcal{B}}^\pi$. We introduce the the bound in brief here: with high probability $\geq 1 - \delta, |\hat{\mathcal{B}}^\pi \hat{Q} - \boldsymbol{B}^\pi \hat{Q}|(s, a) \leq \frac{C_{r, \boldsymbol{P}, \delta}}{\sqrt{|\boldsymbol{D}(s, a)|}}, \forall s, a \in \boldsymbol{D}$, where $C_{r, \boldsymbol{P}, \delta}$ is a constant relating to the reward function $r(s, a)$, environment dynamic $\boldsymbol{P}(\cdot|s, a)$, and $\delta \in (0.1)$.

**Proposition A.2** *(ACQL bounds the gap between the learned Q-values and true Q-values). Considering the sampling error between the empirical $\hat{\mathcal{B}}^\pi \hat{Q}$ and true Bellman backups $\boldsymbol{B}^\pi \hat{Q}$, with a high probability $\geq 1 - \delta$, the gap between the learned Q-function $\hat{Q}^\pi_{ACQL}$ and the true Q-function $Q^\pi$ satisfies the following inequality:*

$$\forall s \in \boldsymbol{D}, a, \ g(s,a) - err(s,a) \leq \hat{Q}^\pi_{ACQL}(s,a) - Q^\pi(s,a) \leq g(s,a) + err(s,a), \tag{29}$$

*where*

$$g(s,a) = - \left[ (I - \gamma P^\pi)^{-1} \frac{d_\mu \cdot \mu - d_{\pi_\beta} \cdot \hat{\pi}_\beta}{\hat{\pi}_\beta} \right] (s,a), \tag{30}$$

$$err(s,a) = \left[ (I - \gamma P^\pi)^{-1} \frac{C_{r,\boldsymbol{P},\delta} R_{max}}{(1 - \gamma \sqrt{|\boldsymbol{D}|})} \right] (s,a) \geq 0. \tag{31}$$

*1) Thus, if $d_\mu(s,a) = d_{\pi_\beta}(s,a) = \alpha, \forall s \in \boldsymbol{D}, a$, it is the case of CQL, where the $\hat{V}^\pi$ lower-bounds the $V^\pi$ with a large $\alpha$ instead of a point-wise lower-bound for Q-function.*
*2) If $g(s,a) \geq err(s,a), \exists s \in \boldsymbol{D}, a$, with the left inequality, the learned Q-function $\hat{Q}^\pi_{ACQL}$ is more optimistic than the true Q-function $Q^\pi$ in these regions.*
*3) If $g(s,a) \leq -err(s,a), \exists s \in \boldsymbol{D}, a$, with the right inequality, the learned Q-function $\hat{Q}^\pi_{ACQL}$ is more conservative than the true Q-function $Q^\pi$ in these regions.*

Note that the term $err(s,a)$ is a positive value for any state-action pair. Given the bounds in Proposition A.2, instead of only knowing the size relationship (i.e., more or less conservative), we can control the fine-grained range of the gap more precisely by carefully designing $d_\mu(s,a)$ and $d_{\pi_\beta}(s,a)$.

**Proof of Proposition A.2**. In Proposition A.1, we calculate the gap between the learned Q-function $\hat{Q}^\pi_{ACQL}$ and the true Q-function $Q^\pi$, representing the conservative level, without the sampling error between the empirical $\hat{\mathcal{B}}^\pi \hat{Q}$ and true Bellman backups $\boldsymbol{B}^\pi \hat{Q}$. Now we can obtain a more precise bound for the conservative level with the consideration of the sampling error. Following (Auer et al., 2008; Osband et al., 2016; Kumar et al., 2020), we can relate the empirical Bellman backups $\hat{\mathcal{B}}^\pi \hat{Q}$ and true Bellman backups $\boldsymbol{B}^\pi \hat{Q}$ as the following: with high probability $\geq 1 - \delta, \delta \in (0,1)$,

$$\forall Q, s, a \in \boldsymbol{D}, \ |\hat{\mathcal{B}}^\pi \hat{Q} - \boldsymbol{B}^\pi \hat{Q}|(s,a) \leq \frac{C_{r,\boldsymbol{P},\delta} R_{max}}{(1 - \gamma)\sqrt{|\boldsymbol{D}(s,a)|}}, \tag{32}$$

where $R_{max}$ is the upper bound for the reward function (i.e., $|r(s,a)| \leq R_{max}$), $C_{r,\boldsymbol{P},\delta}$ is a constant relating to the reward function $r(s,a)$, environment dynamic $\boldsymbol{P}(\cdot|s,a)$. More detailed proofs of the relationship are provided in (Kumar et al., 2020).

For the right inequality in Equation 29, we reason the fixed point of the Q-function in ACQL as the following:

$$|\hat{\mathcal{B}}^\pi \hat{Q}_{ACQL} - \boldsymbol{B}^\pi \hat{Q}_{ACQL}|(s,a) \leq \frac{C_{r,\boldsymbol{P},\delta} R_{max}}{(1-\gamma)\sqrt{|\boldsymbol{D}(s,a)|}} \tag{33}$$

$$\Rightarrow \hat{\mathcal{B}}^\pi \hat{Q}_{ACQL} \leq \boldsymbol{B}^\pi \hat{Q}_{ACQL} + \frac{C_{r,\boldsymbol{P},\delta} R_{max}}{(1-\gamma)\sqrt{|\boldsymbol{D}(s,a)|}} \tag{34}$$

$$\Rightarrow \hat{\mathcal{B}}^\pi \hat{Q}_{ACQL} - \frac{d_\mu \cdot \mu - d_{\pi_\beta} \cdot \hat{\pi}_\beta}{\hat{\pi}_\beta} \leq \boldsymbol{B}^\pi \hat{Q}_{ACQL} - \frac{d_\mu \cdot \mu - d_{\pi_\beta} \cdot \hat{\pi}_\beta}{\hat{\pi}_\beta} + \frac{C_{r,\boldsymbol{P},\delta} R_{max}}{(1-\gamma)\sqrt{|\boldsymbol{D}(s,a)|}} \tag{35}$$

$$\Rightarrow \hat{Q}_{ACQL}^\pi \leq \boldsymbol{B}^\pi \hat{Q}_{ACQL} - \frac{d_\mu \cdot \mu - d_{\pi_\beta} \cdot \hat{\pi}_\beta}{\hat{\pi}_\beta} + \frac{C_{r,\boldsymbol{P},\delta} R_{max}}{(1-\gamma)\sqrt{|\boldsymbol{D}(s,a)|}} \tag{36}$$

$$\Rightarrow \hat{Q}_{ACQL}^\pi \leq (r + \gamma P^\pi \hat{Q}_{ACQL}^\pi) - \frac{d_\mu \cdot \mu - d_{\pi_\beta} \cdot \hat{\pi}_\beta}{\hat{\pi}_\beta} + \frac{C_{r,\boldsymbol{P},\delta} R_{max}}{(1-\gamma)\sqrt{|\boldsymbol{D}(s,a)|}} \tag{37}$$

$$\Rightarrow \hat{Q}_{ACQL}^\pi \leq (I - \gamma P^\pi)^{-1}\left[r - \frac{d_\mu \cdot \mu - d_{\pi_\beta} \cdot \hat{\pi}_\beta}{\hat{\pi}_\beta} + \frac{C_{r,\boldsymbol{P},\delta} R_{max}}{(1-\gamma)\sqrt{|\boldsymbol{D}(s,a)|}}\right] \tag{38}$$

$$\Rightarrow \hat{Q}_{ACQL}^\pi \leq Q^\pi - (I - \gamma P^\pi)^{-1}\frac{d_\mu \cdot \mu - d_{\pi_\beta} \cdot \hat{\pi}_\beta}{\hat{\pi}_\beta} + (I - \gamma P^\pi)^{-1}\frac{C_{r,\boldsymbol{P},\delta} R_{max}}{(1-\gamma)\sqrt{|\boldsymbol{D}(s,a)|}} \tag{39}$$

$$\Rightarrow \hat{Q}_{ACQL}^\pi - Q^\pi \leq g(s,a) + err(s,a). \tag{40}$$

For the left inequality in Equation 29, we have the similar process as the following:

$$|\hat{\mathcal{B}}^\pi \hat{Q}_{ACQL} - \boldsymbol{B}^\pi \hat{Q}_{ACQL}|(s,a) \leq \frac{C_{r,\boldsymbol{P},\delta} R_{max}}{(1-\gamma)\sqrt{|\boldsymbol{D}(s,a)|}} \tag{41}$$

$$\Rightarrow \hat{\mathcal{B}}^\pi \hat{Q}_{ACQL} \geq \boldsymbol{B}^\pi \hat{Q}_{ACQL} - \frac{C_{r,\boldsymbol{P},\delta} R_{max}}{(1-\gamma)\sqrt{|\boldsymbol{D}(s,a)|}} \tag{42}$$

$$\Rightarrow \hat{\mathcal{B}}^\pi \hat{Q}_{ACQL} - \frac{d_\mu \cdot \mu - d_{\pi_\beta} \cdot \hat{\pi}_\beta}{\hat{\pi}_\beta} \geq \boldsymbol{B}^\pi \hat{Q}_{ACQL} - \frac{d_\mu \cdot \mu - d_{\pi_\beta} \cdot \hat{\pi}_\beta}{\hat{\pi}_\beta} - \frac{C_{r,\boldsymbol{P},\delta} R_{max}}{(1-\gamma)\sqrt{|\boldsymbol{D}(s,a)|}} \tag{43}$$

$$\Rightarrow \hat{Q}_{ACQL}^\pi \geq \boldsymbol{B}^\pi \hat{Q}_{ACQL} - \frac{d_\mu \cdot \mu - d_{\pi_\beta} \cdot \hat{\pi}_\beta}{\hat{\pi}_\beta} - \frac{C_{r,\boldsymbol{P},\delta} R_{max}}{(1-\gamma)\sqrt{|\boldsymbol{D}(s,a)|}} \tag{44}$$

$$\Rightarrow \hat{Q}_{ACQL}^\pi \geq (r + \gamma P^\pi \hat{Q}_{ACQL}^\pi) - \frac{d_\mu \cdot \mu - d_{\pi_\beta} \cdot \hat{\pi}_\beta}{\hat{\pi}_\beta} - \frac{C_{r,\boldsymbol{P},\delta} R_{max}}{(1-\gamma)\sqrt{|\boldsymbol{D}(s,a)|}} \tag{45}$$

$$\Rightarrow \hat{Q}_{ACQL}^\pi \geq (I - \gamma P^\pi)^{-1}\left[r - \frac{d_\mu \cdot \mu - d_{\pi_\beta} \cdot \hat{\pi}_\beta}{\hat{\pi}_\beta} - \frac{C_{r,\boldsymbol{P},\delta} R_{max}}{(1-\gamma)\sqrt{|\boldsymbol{D}(s,a)|}}\right] \tag{46}$$

$$\Rightarrow \hat{Q}_{ACQL}^\pi \geq Q^\pi - (I - \gamma P^\pi)^{-1}\frac{d_\mu \cdot \mu - d_{\pi_\beta} \cdot \hat{\pi}_\beta}{\hat{\pi}_\beta} - (I - \gamma P^\pi)^{-1}\frac{C_{r,\boldsymbol{P},\delta} R_{max}}{(1-\gamma)\sqrt{|\boldsymbol{D}(s,a)|}} \tag{47}$$

$$\Rightarrow \hat{Q}_{ACQL}^\pi - Q^\pi \geq g(s,a) - err(s,a). \tag{48}$$

**Proposition A.3** *(The conservative level compared to CQL). For any $\mu$ with $\operatorname{supp}\mu \subset \operatorname{supp}\hat{\pi}_\beta$, given the Q-function learned from CQL is $\hat{Q}_{CQL}^\pi(s,a) = Q^\pi - \alpha\frac{\mu-\pi_\beta}{\pi_\beta}$, similiar as in A.1, the conservative level of ACQL compared to CQL can be controlled at least at three different levels according to different conditions:*

*1) Control over the Q-values. The learned Q-function $\hat{Q}^\pi$ is less conservative than the CQL Q-function $\hat{Q}_{CQL}^\pi$ point-wise, if:*

$$\forall s \in \boldsymbol{D}, a, \ \frac{(\alpha - d_\mu)\mu - (\alpha - d_{\pi_\beta})\pi_\beta}{\pi_\beta} \geq 0. \tag{49}$$

*2) Control over the V-values.* *The excepted values of learned Q-function $\hat{Q}^\pi$ is less conservative than excepted values of the true Q-function $Q^\pi$, if:*

$$\forall s \in \boldsymbol{D}, \sum_a \frac{(\alpha - d_\mu)\mu - (\alpha - d_{\pi_\beta})\pi_\beta}{\pi_\beta} \geq 0. \tag{50}$$

*3) Control over the empirical MDP.* *The learned Q-function $\hat{Q}^\pi$ is less conservative over the empirical MDP, if:*

$$\sum_{s \in \boldsymbol{D}} \sum_a \frac{(\alpha - d_\mu)\mu - (\alpha - d_{\pi_\beta})\pi_\beta}{\pi_\beta} \geq 0. \tag{51}$$

**Proof of Proposition A.3**. As shown in CQL (Kumar et al., 2020), we first recap the optimization problem of CQL as the following:

$$\hat{Q}_{CQL}^{k+1} \leftarrow \min_Q \alpha \left( \mathbb{E}_{s\sim\boldsymbol{D},a\sim\mu(a|s)} \left[ Q(s,a) \right] - \mathbb{E}_{s\sim\boldsymbol{D},a\sim\pi_\beta(a|s)} \left[ Q(s,a) \right] \right)$$

$$+ \frac{1}{2} \mathbb{E}_{s,a,s'\sim\boldsymbol{D}} \left[ \left( Q(s,a) - \boldsymbol{B}^\pi \hat{Q}^k(s,a) \right)^2 \right]. \tag{52}$$

We can observa that CQL is the a special case of ACQL where the adaptive weight functions $d_\mu(s,a)$ and $d_{\pi_\beta}(s,a)$ are both constant $\alpha$. By setting the derivative of Equation 52 to 0, we can obtain the form of the resulting Q-function $\hat{Q}^{k+1}$ in CQL:

$$\forall s \in \boldsymbol{D}, a, \frac{\partial \hat{Q}_{CQL}^{k+1}}{\partial Q} = 0 \tag{53}$$

$$\Rightarrow \alpha \cdot \mu - \alpha \cdot \pi_\beta + \pi_\beta \cdot (Q - \boldsymbol{B}^\pi \hat{Q}^k) = 0 \tag{54}$$

$$\Rightarrow \hat{Q}_{CQL}^{k+1} = \boldsymbol{B}^\pi \hat{Q}^k - \alpha \frac{\mu - \pi_\beta}{\pi_\beta}. \tag{55}$$

Similar to the proof of Proposition A.1, we calculate the difference between the Q-values of ACQL and CQL in Equations (28) and (55) as the following:

$$\forall s \in \boldsymbol{D}, a, \ \hat{Q}_{ACQL}^{k+1} - \hat{Q}_{CQL}^{k+1} \tag{56}$$

$$= \boldsymbol{B}^\pi \hat{Q}^k - \frac{d_\mu \cdot \mu - d_{\pi_\beta} \cdot \pi_\beta}{\pi_\beta} - \boldsymbol{B}^\pi \hat{Q}^k + \alpha \frac{\mu - \pi_\beta}{\pi_\beta} \tag{57}$$

$$= \frac{(\alpha - d_\mu)\mu - (\alpha - d_{\pi_\beta})\pi_\beta}{\pi_\beta}. \tag{58}$$

If we want to learn a less conservative Q-function than CQL in ACQL, we need to make the difference in Equation 58 greater than 0, as shown in Equation 49. If we want to relax the conservative level like requiring to control over the V-values or the empirical MDP, we can easily relax the Equation 58 to the integration over each state or the whole empirical MDP as shown in Equations (50) and (51) respectively. For the conditions where we want to make the learned Q-function $\hat{Q}_{ACQL}^\pi$ is more conservative than the CQL Q-function $\hat{Q}_{CQL}^\pi$, we can easily replace the "$\geq$" to "$\leq$" in Proposition A.3.

**Lemma A.1** *For $x > 0$, $\ln x \leq x - 1$. $\ln x = x - 1$ if and only if $x = 1$.*

**Proof of Lemma A.1**.

Suppose $f(x) = \ln x - x + 1$, then $f'(x) = \frac{1-x}{x}$. It is easy to know that $f(x) \uparrow$ over $(0, 1)$ and $f(x) \downarrow$ over $(1, +\infty)$. Then $f(x) \leq f(1) = 0$.

## B EXPERIMENTAL DETAILS

**Software.** We run our experiments with the following packages and softwares:

- d4rl 1.1
- Python 3.8.13
- Pytorch 1.10.0+cu111
- Gym 0.23.1
- MuJoCo 2.1.5
- mujoco-py 2.1.2.14
- dm-control 1.0.2
- numpy 1.22.3
- h5py 3.6.0

Table 6: Implementation Details of ACQL.

|  | Hyperparameter | Value |
|---|---|---|
| ACQL Hyperparameter | Critic learning rate | 3e-4 |
|  | Actor learning rate | 1e-5 |
|  | Target update rate | 5e-3 |
|  | Optimizer | Adam (Kipf & Welling, 2016) |
|  | Batch size | 256 |
|  | Discount factor | 0.99 |
|  | Lagrange for $\alpha$ | False |
|  | $\alpha$ | 20.0 |
|  | Training steps | 1.5 M |
|  | Evaluation frequency | 1000 |
|  | Evaluation episodes | 10 |
|  | Weight function learning rate | 3e-4 |
|  | Behavior policy training steps | 1.0 M |
| ACQL Network Architectures | Actor type | Tanh & Gaussian |
|  | Actor hidden layers | 3 |
|  | Actor hidden dim | [256, 256, 256] |
|  | Actor activation function | ReLU (Agarap, 2018) |
|  | Critic hidden layers | 3 |
|  | Critic hidden dim | [256, 256, 256] |
|  | Critic activation function | ReLU |
|  | Weight function hidden layers | 3 |
|  | Weight function hidden dim | [256, 256, 256] |
|  | Weight function ouput dim | 2 |
|  | Weight function activation function | ReLU |

**Implementation Details of ACQL.** In ACQL, we represent two adaptive weight functions $d_\mu(s, a)$ and $d_{\pi_\beta}(s, a)$ by one neural network, which has the same network architecture as the Q-function but the output dimension is 2. During each gradient descent step, we train the weight network, the Q networks and the policy networks in turn. We list all the hyperparameters and network architectures of ACQL in Table 6.

## C    ADDITIONAL EXPERIMENTS

### C.1    COMPARISONS TO OFFLINE RL BASELINES ON GYM-MUJOCO-V2 DATASETS

Aiming to provide a more comprehensive comparison, we compared ACQL to many state-of-the-art model-free algorithms including behavioral cloning (BC), 10%BC, Decision Transformer (DT) (Chen et al., 2021), AWAC (Nair et al., 2020), Onestep RL (Brandfonbrener et al., 2021), TD3+BC (Fujimoto & Gu, 2021), IQL (Kostrikov et al., 2021a) and CQL (Kumar et al., 2020) on the version-2 dataset. For the sake of fair comparisons, we directly reported the results of all baselines from the D4RL whitepaper (Fu et al., 2020) and their original papers.

Table 7 shows the normalized returns of all 15 Gym-MuJoCo tasks on the version 2 datasets. Due to most baselines did not report their results on "-expert" datasets, we conducted experiments on other 4 types of datasets. We can observe that ACQL consistently outperforms other baselines on all three environments, specifically on the Hopper environment, i.e., achieving normalized returns 302.9 for hopper-sum compared to 268.2 from CQL which is the second rank. From the perspective of the dataset types, ACQL is a very balanced algorithm that achieves excelling results on all kinds of dataset with expert, medium and random data. Furthermore, we find that ACQL can achieve higher performance than other baselines when there are more medium and random data in datasets, we argue it is because ACQL can adaptively calculate the weights for data with different qualities and thus learn a better policy. For instance, while the best result on hopper-random-v2 of other baselines is 9.6 from AWAC, the result of ACQL is 31.4, which is more than 3 times better.

Table 7: Normalized results on D4RL Gym-MuJoCo environments.

| Task Name | BC | 10%BC | DT | AWAC | OnestepRL | TD3+BC | IQL | CQL | ACQL |
|---|---|---|---|---|---|---|---|---|---|
| halfcheetah-m-e-v2 | 55.2 | 92.9 | 86.8 | 36.8 | **93.4** | 90.7 | 86.7 | 91.6 | 87.9 |
| halfcheetah-m-r-v2 | 36.6 | 40.6 | 36.6 | 40.5 | 38.1 | 44.6 | 44.2 | 45.5 | **46.3** |
| halfcheetah-m-v2 | 42.6 | 42.5 | 42.6 | 37.4 | **48.4** | 48.3 | 47.4 | 44.0 | **48.4** |
| halfcheetah-r-v2 | 2.3 | 2.0 | 2.2 | 2.2 | 6.9 | 11.0 | - | 18.6 | **25.9** |
| halfcheetah-sum | 136.7 | 178.0 | 168.2 | 116.9 | 186.8 | 194.6 | 178.3 | 199.7 | **208.5** |
| hopper-m-e-v2 | 52.5 | **110.9** | 107.6 | 80.9 | 103.3 | 98.0 | 91.5 | 105.4 | 82.2 |
| hopper-m-r-v2 | 18.1 | 75.9 | 82.7 | 37.2 | **97.5** | 60.9 | 94.7 | 95.0 | **97.9** |
| hopper-m-v2 | 52.9 | 56.9 | 67.6 | **72.0** | 59.6 | 59.3 | 66.3 | 58.5 | **91.4** |
| hopper-r-v2 | 4.8 | 4.1 | 7.5 | **9.6** | 7.8 | 8.5 | - | 9.3 | **31.4** |
| hopper-sum | 128.3 | 247.8 | 265.4 | 199.7 | 268.2 | 226.7 | 252.5 | 268.2 | **302.9** |
| walker-m-e-v2 | 107.5 | 109.0 | 108.1 | 42.7 | **113.0** | 110.1 | 109.6 | 108.8 | 108.3 |
| walker-m-r-v2 | 26.0 | 62.5 | 66.6 | 27.0 | 49.5 | **81.8** | 73.9 | 77.2 | **85.5** |
| walker-m-v2 | 75.3 | 75.0 | 74.0 | 30.1 | 81.8 | **83.7** | 78.3 | 72.5 | **84.9** |
| walker-r-v2 | 1.7 | 1.7 | 2.0 | 5.1 | **6.1** | 1.6 | - | 2.5 | 5.2 |
| walker-sum | 210.5 | 248.2 | 250.7 | 104.9 | 250.4 | **277.2** | 261.8 | 261.0 | **283.9** |

### C.2    COMPARISONS AMONG DIFFERENT CONSERVATIVE LEVELS

To demonstrate more intuitively that ACQL can adaptively control the conservative level, we compared ACQL to CQL with different $\alpha$ values ranging from $1, 2, 5, 10, 20$. The different $\alpha$ values represent different conservative levels and a higher $\alpha$ means more conservative.

Figure 4 plotted the learning curves of ACQL and CQL with 5 kinds of $\alpha$. From the Figure 4, one trend we can clearly observe is that CQL with a higher conservative level can usually achieve higher performance on expert datasets as opposed to that a lower conservative level is more effective for random datasets. For instance, the CQL agent with $\alpha = 20$ (green line) delivered rapidly increasing results as the training epochs increased on the halfcheetah-expert dataset (top left corner), while hovered at the bottom of the plot on the halfcheetah-random dataset (top right corner). Moreover, it is difficult for CQL to achieve satisfactory performance on all kinds of dataset with a fixed $\alpha$. It is exactly the issue that the ACQL focuses on. Since ACQL generates adaptive weight for each state-action pair and control the conservative level in a more fine-grained way, ACQL can achieve balanced and state-of-the-art results for different dataset types. In addition, in the plot of hopper-medium-replay task (2th row, 2th column), ACQL outperformed CQL agents by a large margin,

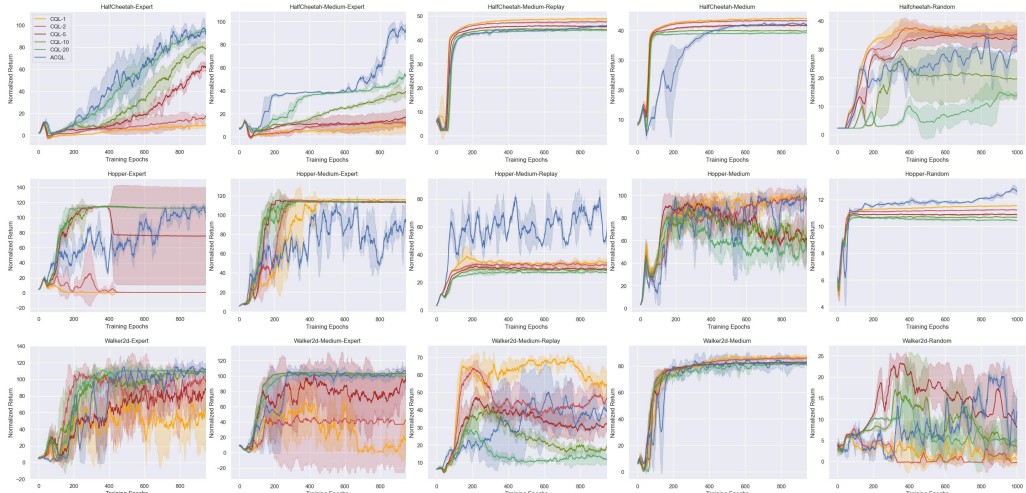

Figure 4: Learning curves comparing to CQL with different conservative levels.

Table 8: Normalized results on D4RL Gym-MuJoCo environments.

| Task Name | CQL-1 | CQL-2 | CQL-5 | CQL-10 | CQL-20 | ACQL |
|---|---|---|---|---|---|---|
| halfcheetah-e-v0 | 9.8 | 18.7 | 67.1 | 84.5 | 96.3 | **97.7** |
| halfcheetah-m-e-v0 | 9.7 | 12.0 | 17.4 | 40.0 | 53.9 | **93.2** |
| halfcheetah-m-r-v0 | **48.9** | 47.5 | 45.7 | 44.4 | 43.9 | 46.4 |
| halfcheetah-m-v0 | **43.9** | 43.3 | 41.7 | 39.9 | 39.2 | 42.8 |
| halfcheetah-r-v0 | **36.8** | 35.3 | 33.6 | 19.7 | 13.7 | 31.3 |
| halfcheetah-sum | 149.1 | 156.8 | 205.5 | 228.5 | 247.0 | **311.4** |
| hopper-e-v0 | 28.2 | 48.5 | 75.6 | 112.6 | 112.9 | **116.3** |
| hopper-m-e-v0 | **116.1** | 114.1 | 112.9 | 113.5 | 113.4 | 111.8 |
| hopper-m-r-v0 | 33.3 | 32.1 | 29.5 | 29.1 | 27.1 | **77.3** |
| hopper-m-v0 | **99.8** | 93.3 | 51.7 | 56.5 | 46.4 | 85.8 |
| hopper-r-v0 | 11.5 | 11.2 | 10.9 | 10.7 | 10.4 | **12.7** |
| hopper-sum | 288.9 | 299.2 | 280.6 | 322.4 | 310.2 | **403.9** |
| walker-e-v0 | 44.5 | 103.0 | 84.4 | 108.9 | 109.7 | **116.1** |
| walker-m-e-v0 | 21.7 | 37.1 | 86.9 | **103.5** | 101.6 | 94.5 |
| walker-m-r-v0 | **56.1** | 43.8 | 30.0 | 19.9 | 11.4 | 45.2 |
| walker-m-v0 | **86.9** | 85.3 | 82.9 | 82.2 | 81.1 | 81.8 |
| walker-r-v0 | 0.5 | -0.2 | 8.3 | 3.5 | 4.9 | **11.8** |
| walker-sum | 209.7 | 269.0 | 292.5 | 318.0 | 308.7 | **349.4** |
| all-sum | 647.7 | 725.0 | 778.6 | 868.9 | 865.9 | **1064.7** |

showing the significance of adaptive weight for each data sample instead of using the same weight $\alpha$.

Table 8 showed the quantitative results of ACQL and CQL with 5 kinds of $\alpha$. From the Table 8, one trend we can clearly observe is that CQL with a higher conservative level can usually achieve higher performance on expert datasets as opposed to that a lower conservative level is more effective for random datasets. For instance, the CQL agent with $\alpha = 20$ delivered high performance value of 96.3 on the halfcheetah-expert dataset, while dropped its performance dramatically to 13.7 on the halfcheetah-random dataset. Moreover, it is difficult for CQL to achieve satisfactory performance on all kinds of dataset with a fixed $\alpha$. It is exactly the issue that the ACQL focuses on. Since ACQL generates adaptive weight for each state-action pair and control the conservative level in a more fine-grained way, ACQL can achieve balanced and state-of-the-art results for different dataset types. In addition, on the hopper-medium-replay task, ACQL outperformed CQL agents by a large margin,

Table 9: Normalized results on D4RL AntMaze environments.

| Task Name | BC | 10%BC | DT | AWAC | OnestepRL | TD3+BC | IQL | CQL | ACQL |
|---|---|---|---|---|---|---|---|---|---|
| antmaze-large-diverse-v0 | 0.0 | 6.0 | 0.0 | 1.0 | 0.0 | 0.0 | **47.5** | 14.9 | 0.0 |
| antmaze-large-play-v0 | 0.0 | 0.0 | 0.0 | 0.0 | 0.0 | 0.2 | **39.6** | 15.8 | 0.0 |
| antmaze-medium-diverse-v0 | 0.0 | 9.8 | 0.0 | 0.7 | 0.0 | 3.0 | **70.0** | 53.7 | 0.0 |
| antmaze-medium-play-v0 | 0.0 | 5.4 | 0.0 | 0.0 | 0.3 | 10.6 | **71.2** | 61.2 | 22.2 |
| antmaze-umaze-diverse-v0 | 45.6 | 50.2 | 53.0 | 49.3 | 60.7 | 71.4 | 62.2 | **84.0** | 76.8 |
| antmaze-umaze-v0 | 54.6 | 62.8 | 59.2 | 56.7 | 64.3 | 78.6 | 87.5 | 74.0 | **94.9** |

showing the significance of adaptive weight for each data sample instead of using the same weight $\alpha$.

### C.3 COMPARISONS TO OFFLINE RL BASELINES ON ANTMAZE TASKS

The AntMaze tasks mimic real-world robotic navigation tasks that aim to control an "Ant" quadruped robot to reach the goal location from a start location with only 0-1 sparse rewards given. Aiming to provide a more comprehensive comparison, we compared ACQL to many state-of-the-art model-free algorithms including behavioral cloning (BC), 10%BC, Decision Transformer (DT) (Chen et al., 2021), AWAC (Nair et al., 2020), Onestep RL (Brandfonbrener et al., 2021), TD3+BC (Fujimoto & Gu, 2021), IQL (Kostrikov et al., 2021a) and CQL (Kumar et al., 2020). For the sake of fair comparisons, we directly reported the results of all baselines from the D4RL whitepaper (Fu et al., 2020) and their original papers.

Table 9 shows the normalized returns of all 6 AntMaze tasks. We can observe that ACQL delivers state-of-the-art performance on umaze-diverse-v0 and umaze-v0 datasets, demonstrating its effectiveness on small mazes. However, while IQL can achieve better performance on larger mazes like "-large" and "-medium" tasks, ACQL fails in these scenarios. The experimental results show that it is more difficult for ACQL to tell whether the state-action pairs are good or bad when only 0-1 sparse rewards are given.

### C.4 ADDITIONAL ABLATION STUDY

Besides the main results of comparison to other baselines and a detailed comparison to different conservative levels, we also conducted extensive ablation studies for ACQL to study the effect of the proposed losses. We reported the quantitative results on the all Gym-MuJoCo environments including 15 kinds of datasets in Tables 10, 11 and 12.

Table 10: Ablation Study on D4RL Gym-MuJoCo HalfCheetah-v0 environments in terms of normalized results.

| | $\mathcal{L}_{cl\_true}$ | $\mathcal{L}_{cl\_cql}$ | $\mathcal{L}_{mono}$ | $\mathcal{L}_{pos}$ | halfcheetah-e | halfcheetah-m-e | halfcheetah-m-r | halfcheetah-m | halfcheetah-r |
|---|---|---|---|---|---|---|---|---|---|
| 1 | ✓ | | | | 9.9 | 15.2 | 28.5 | 16.4 | 5.4 |
| 2 | ✓ | | | ✓ | 57.8 | 23.9 | 39.9 | 33.5 | 24.1 |
| 3 | | ✓ | | | 8.8 | 22.5 | 47.6 | **47.0** | 26.9 |
| 4 | | ✓ | | ✓ | 6.7 | 19.9 | **49.6** | 46.4 | 9.8 |
| 5 | ✓ | ✓ | | | 34.7 | 47.7 | 38.3 | 38.1 | 21.4 |
| 6 | ✓ | ✓ | | ✓ | 42.3 | 75.3 | 42.5 | 43.1 | 23.5 |
| 7 | ✓ | ✓ | ✓ | | 27.1 | 37.7 | 44.3 | 40.3 | 31.2 |
| 8 | ✓ | ✓ | ✓ | ✓ | **97.7** | **93.2** | 46.4 | 42.8 | **31.3** |

Table 11: Ablation Study on D4RL Gym-MuJoCo Hopper-v0 environments in terms of normalized results.

| | $\mathcal{L}_{cl\_true}$ | $\mathcal{L}_{cl\_cql}$ | $\mathcal{L}_{mono}$ | $\mathcal{L}_{pos}$ | hopper-e | hopper-m-e | hopper-m-r | hopper-m | hopper-r |
|---|---|---|---|---|---|---|---|---|---|
| 1 | ✓ | | | | 21.7 | 8.3 | 33.1 | 52.8 | 10.6 |
| 2 | ✓ | | | ✓ | 31.8 | 31.3 | 36.6 | 51.9 | **31.7** |
| 3 | | | ✓ | | 21.5 | 31.5 | 38.2 | 58.4 | 24.3 |
| 4 | | | ✓ | ✓ | 21.9 | 36.5 | 70.1 | **93.5** | 12.9 |
| 5 | ✓ | ✓ | | | 34.8 | 85.4 | 66.9 | 41.7 | 12.5 |
| 6 | ✓ | ✓ | | ✓ | 61.9 | 74.5 | **95.4** | 70.9 | 11.7 |
| 7 | ✓ | ✓ | ✓ | | 114.4 | **112.4** | 71.5 | 56.3 | 10.7 |
| 8 | ✓ | ✓ | ✓ | ✓ | **116.3** | 111.8 | 77.3 | 85.8 | 12.7 |

Table 12: Ablation Study on D4RL Gym-MuJoCo Walker environments in terms of normalized results.

| | $\mathcal{L}_{cl\_true}$ | $\mathcal{L}_{cl\_cql}$ | $\mathcal{L}_{mono}$ | $\mathcal{L}_{pos}$ | walker-e | walker-m-e | walker-m-r | walker-m | walker-r |
|---|---|---|---|---|---|---|---|---|---|
| 1 | ✓ | | | | 23.3 | 32.4 | 22.5 | 9.3 | 18.2 |
| 2 | ✓ | | | ✓ | 93.7 | 38.5 | 30.4 | 27.3 | 11.9 |
| 3 | | | ✓ | | 11.6 | 28.3 | **64.8** | 20.2 | 17.5 |
| 4 | | | ✓ | ✓ | 55.1 | 22.4 | 29.7 | **82.8** | 3.5 |
| 5 | ✓ | ✓ | | | 6.3 | **113.5** | 63.4 | 13.3 | **21.7** |
| 6 | ✓ | ✓ | | ✓ | 110.2 | 55.5 | 21.4 | 62.2 | 10.7 |
| 7 | ✓ | ✓ | ✓ | | 102.9 | 81.1 | 28.5 | 81.1 | 11.2 |
| 8 | ✓ | ✓ | ✓ | ✓ | **116.1** | 94.5 | 45.2 | 81.8 | 11.8 |

