# OpenReview forum: "ACQL: An Adaptive Conservative Q-Learning Framework for Offline Reinforcement Learning"
_ICLR.cc/2023/Conference — Submitted to ICLR 2023_

### Official Review · Reviewer_KtHP · 2022-10-24

**Confidence:** 4
**Correctness:** 4
**Technical Novelty And Significance:** 2
**Empirical Novelty And Significance:** 2
**Recommendation:** 5

**Clarity, Quality, Novelty And Reproducibility:**

Clarity: The paper is very clearly written and easy to follow.
Quality: There does not seem to be a wrong argument in the paper, and extensive experiments support the algorithm. However, the algorithm is somewhat ad-hoc and not well supported by the theoretical results.
Originality: This is a paper about souped-up CQL, so it is not that original. The theoretical results also use the proof methods used in CQL.

**Strength And Weaknesses:**

Strength
- performance improvement over CQL
- paper is clearly written and easy to follow
- extensive experiments on many different benchmarks including ablation study

Weakness
- weak support on "monotonicity"
While it is interesting to have proposition 4.1~4.2 and propose weight functions having such characteristics, the definition 4.1 is not well supported. Even in the case where we have $Q^*$, using the weight functions following eq 6, 7 will not recover an optimal policy. If we have some other choice, e.g. weight that is proportional to exponential of $Q$s, we may get some other intuitions on the algorithm by relating it to AWAC or etc.

- not well-motivated transition quality measurements
While the authors suggested using $Q^*$ as a weights, the suggested transition quality measurements are too irrelevant to $Q^*$, even considering that $Q^*$ cannot be computed. There are number of choices that are not explained, e.g.
1. averaging normalized reward and return: it is not a normal choice to average between reward and the return; if we wanted to make a bit more myopic measure than the return, I may just increase the discount factor for this. This choice is too ad-hoc and will be very suboptimal in some specific cases.
2. using the behavior $Q$ instead of $Q^*$: using a simple return $g$ corresponds to using a behavior $Q$ function. It does not matches the above motivation of using $Q^*$ as a weight function.

- about the positivity
I think using the positivity objective (19) won't ensure the model has positive output. What is the reason of not using a positive transformation at the output of the neural network? e.g. softplus

**Summary Of The Paper:**

This paper proposes ACQL, which weights the regularization term used in CQL to improve performance. By using a weight function that is based on heuristics, ACQL regularizes each state action differently according to the weight. In the experiment, the paper shows the results on various benchmarks that ACQL can improve on CQL.

**Summary Of The Review:**

The paper is overall clearly written, and there may be some readers interested in reading the paper. However, in my opinion, the theoretical results are weak, and the algorithm is not well supported by the theory. Despite the extensive evaluation of the algorithm, considering the low originality of ACQL, I am more leaning toward rejection.

---

> ### Author Response · Authors · 2022-11-17
> **Response to Reviewer KtHP**
>
> Thanks a lot for coordinating the review for our paper. We believe that based on the current code status and scoring system of different benchmark versions, we have done the best combination to produce fair results. It took us some time to set up additional experiments and please allow us to explain.
>
> (1) Regarding weak support on "monotonicity".
>
> We discuss the motivation for monotonicity in Section 1.
> As shown in Figure 1, different $\alpha$ values lead to huge performance gaps for different kinds of datasets and thus we need a more fine-grained weight to control the conservative level.
> Furthermore, for "relatively good" state-action pairs, we need to raise their Q-values while suppressing the Q-values of "relatively bad" state-action pairs.
> For the effect of the monotonicity loss, we have built ablation studies in Section 6.3 and in Appendix C.2 (old submission), and Appendix C.4 (new submission).
> From the results of these ablation studies, we can see the monotonicity loss contributes to the final results.
>
> We think some other choices like exponential of $Qs$ from AWAC may also work very well.
> However, one potential issue of using exponential $Qs$ is the Q-values explosion, which is the main challenge in Offline DRL.
> Note that the training losses currently used for the adaptive weight function in Equation 20 are not related to Q-values.
> If we use the exponential of $Qs$ to main the monotonicity, once the Q-values of some state-action pairs are overestimated, the learned adaptive weight will also be overestimated (i.e., the state-action pairs being good!), leading to that Q-values are overestimated in turn.
>
>
> (2) Regarding not well-motivated transition quality measurements
>
> 2.1 averaging normalized reward and return
>
> When the $Q^*$ cannot be computed and the normal Q-function is not reliable in offline setting, the rest choice is the single-step reward and MC return of each transition.
> Thus we both consider the single-step reward and MC return to calculate the relative transition quality $m(s,a)$.
> Equation 9 just looks complicated, but the goal is actually very simple and is to get its range from 0 to 1.
> So we can easily shift and scale its value to match the magnitude of any variables suitable for different datasets.
>
>
> 2.2 using the behavior $Q$ instead of $Q^*$
>
> When we cannot compute the $Q^*$, we think the only choice is to leverage the single-step rewards and MC returns.
> The single-step rewards and MC returns are unbiased and thus we think it can be an approximation of $Q^*$ or at least an initial values.
>
> We believe that the results of ACQL on so many different domains and datasets demonstrate that the proposed transition quality function can work well.
> Of course, the optimal transition quality may not be the best one, and we also did not claim it is the best solution.
> As we discussed in Section 5.1, we think it does have some room for improvement, for example, in combination with other methods including the "upper envelope" in BAIL[1] and uncertainty estimation in [2] has the potential to be incorporated in ACQL.
> In this work, we mainly focus on the ACQL framework that can control the conservative level more adaptively and thus does not make the algorithm too complex.
> How to combinate ACQL with other methods are out of the scope of this paper and we left it to future work.
>
>
>
> (3) Regarding positivity objective (19) won't ensure the model has positive output.
>
> We don't expect that we can use a positive transformation like softplus at the output of the neural network and we think it is a good idea to reduce the optimization objects.
> In the current ACQL, the positivity objective (19) also worked well.
> The final results and ablation studies of ACQL on so many different domains and datasets demonstrate that the positivity objective (19) contributes to the final results.

---

### Official Review · Reviewer_3P64 · 2022-10-25

**Confidence:** 4
**Correctness:** 3
**Technical Novelty And Significance:** 3
**Empirical Novelty And Significance:** 2
**Recommendation:** 5

**Clarity, Quality, Novelty And Reproducibility:**

The paper is overall clear, well-written, and original. The paper has a relatively high theoretical quality but low in its empirical studies.

**Strength And Weaknesses:**

Strengths:

Overall, the paper is well-written, well-motivated, and easy to follow. The paper has studied an important and interesting problem of the famous conservative learning methods for offline reinforcement learning, i.e., the over-conservative value estimation. Theoretically, the proposed ACQL addresses the issue of over-conservative value estimation.

Weaknesses:

The main weakness of this paper is the insufficient empirical results.

1/ The results presented in Table 1 are susceptible. According to the authors, the results are directly taken from the original paper of baselines or the D4RL whitepaper [1]. However, the reported results of CQL are significantly worse than the ones discussed in IQL [2]. For example, in the submitted manuscript, CQL has a normalized return of 26.7 on walker2d-medium-replay, while in IQL paper, CQL has a normalized return of 77.2, which is already higher than the reported ACQL result (45.2). There are 4 such environments where the reported CQL performs worse than the CQL results in the IQL paper, including halfcheetah-m-e, hopper-m-e, hopper-m-r, and walker2d-m-r. The performance gap is caused by the misuse of environments. In gym-locomotion-v0 environments, the performance is generally worse than in the v2 environments. It seems that the reported results of CQL are directly taken from the D4RL whitepaper, where v0 environments are used, while the reported IQL results are run on v2 environments. As a result, the results reported in Table 1 are not comparable. In addition, the submitted manuscript mentions no information about the version of its environment, which further makes it hard to justify the soundness of its empirical results.

2/ No results on antmaze are reported. Antmaze is considered one of the most difficult environments due to its sparse and delayed reward. In addition, low-quality transitions often exist in such environments, which I believe should be an ideal environment to test ACQL against CQL. It would be good to add such results.

3/ In addition to the ablation studies, some intuition should be provided to better understand the influences of these two weights. Although theoretically, ACQL guarantees a monotonic improvement, it remains unclear how such behavior reflects on the learned weights. It would be great to visualize the relationship between the learned weight and the quality of transitions, i.e., m(s, a).

4/ According to the ablation studies, it seems that ACQL is fairly unstable. Without L_{mono}, ACQL would fail completely and even perform worse than the reported CQL result. How about training only with L_{mono} without other losses?

Besides the above-mentioned issues, another issue is that ACQL has multiple losses to ensure the learned weight is reasonable and stable. How does this compare with directly weighting the samples with a fixed weight? For example, m(s, a)?

References:

[1] Fu, J., Kumar, A., Nachum, O., Tucker, G., & Levine, S. (2020). D4RL: Datasets for deep data-driven reinforcement learning. arXiv preprint arXiv:2004.07219.
[2] Kostrikov, I., Nair, A., & Levine, S. (2021). Offline reinforcement learning with implicit q-learning. International Conference on Learning Representations (ICLR)
[3] Kumar, A., Zhou, A., Tucker, G., & Levine, S. (2020). Conservative q-learning for offline reinforcement learning. Advances in Neural Information Processing Systems (NeurIPS)


**Summary Of The Paper:**

This paper introduces ACQL, adaptive conservative Q-learning for offline reinforcement learning. Different from the standard Q learning framework, ACQL learns a less conservative policy by controlling the weight of each sample with a learnable parameter. The weight is directly associated with the statistics of the dataset and the quality of the transitions. Theoretically, ACQL gives monotonic improvement on the conservative level of CQL constraints. Empirically, the authors have presented improved results on the D4RL benchmark across multiple tasks.

**Summary Of The Review:**

I think the paper is interesting and novel. However, as discussed above, the empirical study needs significant improvements. Thus, I think it is not ready for publication yet.

---

> ### Author Response · Authors · 2022-11-17
> **Response to Reviewer 3P64**
>
> Thanks a lot for coordinating the review for our paper. We believe that based on the current code status and scoring system of different benchmark versions, we have done the best combination to produce fair results. It took us some time to set up additional experiments and please allow us to explain.
>
> (1) Regarding the results presented in Table 1 are susceptible.
>
> As we responded to another reviewer AX8y, in Table 1, the results of DT, OnestepRL, and IQL are based on the mujuco v2 datasets and the rest results including CQL and ACQL are based on v0.
> And we directly report the official results of CQL from the D4RL whitepaper, so we think the result comparisons are fair.
> To say the least, even if CQL is flawed on the v0 datasets, that means ACQL is also flawed because we inherited its code.
> In fact, this comparison is unfair to ACQL because the results of other algorithms are not flawed.
> In addition, the results of DT, OnestepRL, and IQL on v2 dataset are usually better than the result on v0 because some bugs in v0 datasets are fixed in v2 datasets.
> That is also why we choose to report v0 results instead of v2, because we do not want to use v2 better results to compare with v0 results of other algorithms with bugs.
> As we report them directly from the D4RL whitepaper or their original papers, we also do not intentionally suppress the results of other algorithms.
>
> Now we have uploaded a new submission that notes all the versions of the datasets in all Tables.
> For Gym-MuJuCo environments, we provide two Tables (Table 1 in Section 6.1 and Table 7 in Appendix C.1) reporting the results on v0 and v2 respectively.
> We also add the tables to this comment[Gym-MuJuCo v0 and v2 results](https://openreview.net/forum?id=o_HqtIc-oF&noteId=FNkyRJzapi6), please check the new results to see if there is any other question.
>
>
> (2) Regarding no results on antmaze are reported
>
> Now we have uploaded a new submission that includes the experiments on AntMaze tasks in Appendix C.3.
> We also add the tables to this comment as the following:
>
> Table 9:  Normalized results on D4RL AntMaze environments.
> |  Task Name  | BC | 10%BC | DT | AWAC | OnestepRL | TD3+BC | IQL | CQL | ACQL |
> |  ----  | ----  | ----  | ----  | ----  | ----  | ----  | ----  | ----  | ----  |
> | antmaze-large-diverse-v0 | 0.0 | 6.0 | 0.0 | 1.0 | 0.0 | 0.0 | **47.5** | 14.9 | 0.0 |
> | antmaze-large-play-v0  | 0.0 | 0.0 | 0.0 | 0.0 | 0.0 | 0.2 | **39.6** | 15.8 | 0.0 |
> | antmaze-medium-diverse-v0 | 0.0 | 9.8 | 0.0 | 0.7 | 0.0 | 3.0 | **70.0** | 53.7 | 0.0 |
> | antmaze-medium-play-v0 | 0.0 | 5.4 | 0.0 | 0.0 | 0.3 | 10.6 | **71.2** | 61.2 | 22.2 |
> | antmaze-umaze-diverse-v0 | 45.6 | 50.2 | 53.0 | 49.3 | 60.7 | 71.4 | 62.2 | **84.0** | 76.8 |
> | antmaze-umaze-v0  | 54.6 | 62.8 | 59.2 | 56.7 | 64.3 | 78.6 | **87.5** | 74.0 | **94.9** |
>
> We can observe that ACQL delivers state-of-the-art performance on umaze-diverse-v0 and umaze-v0 datasets, demonstrating its effectiveness on small mazes.
> However, while IQL can achieve better performance on larger mazes like "-large" and "-medium" tasks, ACQL fails in these scenarios.
> We think it is because the  0-1 sparse reward in AntMaze tasks is too extreme for ACQL, so it is more difficult to learn a good adaptive weight function, especially for the large maze.
> In contrast, CQL can achieve medium performance on large mazes because its $\alpha$ is set manually.
>
> (3) Regarding visualizing the relationship between the learned weight and the quality of transitions
>
> Now we have uploaded a new submission with the visualization of the adaptive Weights (Figure 3) in Section C.3.
> On the left of Figure 3, we visualize the relationship between the learned weight and the quality of transitions for OOD state-action pairs.
> We can see that as the quality measurements (green and red points) increase, the adaptive weights $d_{\mu}$ (blue and yellow points) decrease, indicating the Q-values of good OOD state-action pairs should be suppressed less.
> In addition, the value of adaptive weights $d_{\mu}\_rand$ (blue points) is larger than $d_{\mu}\_policy$ (yellow points), representing that the Q-values of random actions should be suppressed more since random actions are very likely to be unsafe and lead to bad performance.
> On the right of Figure 3, we visualize the relationship between the learned weight and the quality of transitions for in-dataset state-action pairs.
> As the quality measurements (green points) increase, the adaptive weights $d_{\pi_{\beta}}$ (blue points) also increase, indicating the Q-values of good in-dataset state-action pairs should be higher.
> Note that the monotonicity is not very strict while the trend exists, since we alternatively train the Q-function, policy, and adaptive weight function and perform only one step of gradient descent each time.

---

> > ### Author Response · Authors · 2022-11-17
> > **Response to Reviewer 3P64**
> >
> > (4) Regarding ACQL is fairly unstable and how about training only with L_{mono} without other losses
> >
> > We have also noticed that the current ACQL is not stable in some environments, and we believe that this may be due to the following reasons.
> > 1) We add an extra optimization process for the adaptive weight function.
> > 2) During the training process, the predicted $d_{\mu}$ and $d_{\pi_{\beta}}$ are not smooth enough between two iterations, leading to fluctuations in the calculation of target Q values.
> > We think one possible solution is that we can separate out the training of the adaptive weight function from the Q-function and policy, which is similar to onestep RL[1].
> > Another way is to perform more steps of gradient descent to make the adaptive weight function coverage for each iteration.
> > However, it is computationally heavy.
> > In this work, we mainly focus on the ACQL framework that can control the conservative level more adaptively and thus does not make the algorithm too complex.
> > How to combinate ACQL with other methods to make ACQL more stable are out of the scope of this paper and we left it to future work.
> >
> > For the experiments training only with L_{mono} without other losses, because it took us some time to set up additional experiments on AntMaze tasks and MuJuCo-v2 tasks, we are still running and we will update here once we get the results.
> >
> > (5) Regarding directly weighting the samples with a fixed weight
> >
> > We have also privately used a hand-designed function to directly calculate the adaptive weights and obtained some preliminary results.
> > These results are not as good as the current method, so we did not report them in the paper.
> > The advantage of a hand-designed function is that it is easy to design a function with monotonicity.
> > But the problem with a hand-designed function is that it is difficult to guarantee that it can keep the conservative level within a certain range because the conservative level is also related to current policy $\pi$ which changes during the training process.
> > Of course, we still think that hand-designed functions are also a promising way to improve ACQL and we also left it to future work.
> >
> > [1] Brandfonbrener, David, et al. "Offline rl without off-policy evaluation." Advances in Neural Information Processing Systems 34 (2021): 4933-4946.

---

> > > ### Author Response · Authors · 2022-11-22
> > > **Response to Reviewer 3P64**
> > >
> > > (4) Regarding ACQL is fairly unstable and how about training only with L_{mono} without other losses
> > >
> > > Now we have conducted the ablation studies about training only with $L_{mono}$ without other losses and add the results here since we are not allowed to upload new submission.
> > >
> > > Table 10: Ablation Study on D4RL Gym-MuJoCo HalfCheetah-v0 environments in terms of normalized results.
> > >
> > > |  | $L_{cl_{true}}$ | $L_{cl_{cql}}$ | $L_{mono}$ | $L_{pos}$ | halfcheetah-e | halfcheetah-m-e | halfcheetah-m-r | halfcheetah-m | halfcheetah-r |
> > > |  :----:  | :----:  | :----:  | :----:  | :----:  | :----:  | :----:  | :----:  | :----:  | :----:  |
> > > | 1 | ✓ | - | - | - | 9.9 | 15.2 | 28.5 | 16.4 | 5.4 |
> > > | 2 | ✓ | - | - | ✓ | 57.8 | 23.9 | 39.9 | 33.5 | 24.1 |
> > > | 3 | - | ✓ | - | - | 8.8 | 22.5 | 47.6 | **47.0** | 26.9 |
> > > | 4 | - | ✓ | - | ✓ | 6.7 | 19.9 | **49.6** | 46.4 | 9.8 |
> > > | 5 | ✓ | ✓ | - | - | 34.7 | 47.7 | 38.3 | 38.1 | 21.4 |
> > > | 6 | ✓ | ✓ | - | ✓ | 42.3 | 75.3 | 42.5 | 43.1 | 23.5 |
> > > | 7 | - | - | ✓ | - | 7.8 | 19.5 | 30.9 | 18.6 | 31.0 |
> > > | 8 | ✓ | ✓ | ✓ | - | 27.1 | 37.7 | 44.3 | 40.3 | 31.2 |
> > > | 9 | ✓ | ✓ | ✓ | ✓ | **97.7** | **93.2** | 46.4 | 42.8 | **31.3** |
> > >
> > > Table 11: Ablation Study on D4RL Gym-MuJoCo Hopper-v0 environments in terms of normalized results.
> > >
> > > |  | $L_{cl_{true}}$ | $L_{cl_{cql}}$ | $L_{mono}$ | $L_{pos}$ | hopper-e | hopper-m-e | hopper-m-r | hopper-m | hopper-r |
> > > |  :----:  | :----:  | :----:  | :----:  | :----:  | :----:  | :----:  | :----:  | :----:  | :----:  |
> > > | 1 | ✓ | - | - | - | 21.7 | 8.3 | 33.1 | 52.8 | 10.6 |
> > > | 2 | ✓ | - | - | ✓ | 31.8 | 31.3 | 36.6 | 51.9 | **31.7** |
> > > | 3 | - | ✓ | - | - | 21.5 | 31.5 | 38.2 | 58.4 | 24.3 |
> > > | 4 | - | ✓ | - | ✓ | 21.9 | 36.5 | 70.1 | **93.5** | 12.9 |
> > > | 5 | ✓ | ✓ | - | - | 34.8 | 85.4 | 66.9 | 41.7 | 12.5 |
> > > | 6 | ✓ | ✓ | - | ✓ | 61.9 | 74.5 | **95.4** | 70.9 | 11.7 |
> > > | 7 | - | - | ✓ | - | 28.6 | 64.4 | 20.3 | 55.1 | 10.1 |
> > > | 8 | ✓ | ✓ | ✓ | - | 114.4 | **112.4** | 71.5 | 56.3 | 10.7 |
> > > | 9 | ✓ | ✓ | ✓ | ✓ | **116.3** | 111.8 | 77.3 | 85.8 | 12.7 |
> > >
> > >
> > > Table 12: Ablation Study on D4RL Gym-MuJoCo Walker-v0 environments in terms of normalized results.
> > >
> > > |  | $L_{cl_{true}}$ | $L_{cl_{cql}}$ | $L_{mono}$ | $L_{pos}$ | walker-e | walker-m-e | walker-m-r | walker-m | walker-r |
> > > |  :----:  | :----:  | :----:  | :----:  | :----:  | :----:  | :----:  | :----:  | :----:  | :----:  |
> > > | 1 | ✓ | - | - | - | 23.3 | 32.4 | 22.5 | 9.3 | 18.2 |
> > > | 2 | ✓ | - | - | ✓ | 93.7 | 38.5 | 30.4 | 27.3 | 11.9 |
> > > | 3 | - | ✓ | - | - | 11.6 | 28.3 | **64.8** | 20.2 | 17.5 |
> > > | 4 | - | ✓ | - | ✓ | 55.1 | 22.4 | 29.7 | **82.8** | 3.5 |
> > > | 5 | ✓ | ✓ | - | - | 6.3 | **113.5** | 63.4 | 13.3 | **21.7** |
> > > | 6 | ✓ | ✓ | - | ✓ | 110.2 | 55.5 | 21.4 | 62.2 | 10.7 |
> > > | 7 | - | - | ✓ | - | 17.1 | 43.4 | 29.2 | 17.6 | 8.1 |
> > > | 8 | ✓ | ✓ | ✓ | - | 102.9 | 81.1 | 28.5 | 81.1 | 11.2 |
> > > | 9 | ✓ | ✓ | ✓ | ✓ | **116.1** | 94.5 | 45.2 | 81.8 | 11.8 |
> > >
> > > From the results in the tables above, we can observe that training only with $L_{mono}$ without other losses (row 7) has a significant drop in performance on all environments and all types of datasets.
> > > We argue that it is because the Q-values are not limited in a conservative range and thus even the monotonicity can be maintained, the Q-values of OOD actions can be also erroneously overestimated.

---

> > > > ### Comment · Reviewer_3P64 · 2022-12-06
> > > > **Re response**
> > > >
> > > >
> > > > I thank the authors for their detailed response and efforts to improve this paper with many additional experiments, ablation studies, and visualizations.
> > > >
> > > > The response has addressed some of my concerns. In particular, ACQL produces strong results compared with other baselines on the v2 dataset. The additional ablation studies and visualizations also explain the mechanisms under the hood. However, it seems the weighting strategy of ACQL is too aggressive, and as a result, it failed in the more challenging AntMaze environments.
> > > >
> > > > I want to raise my score to 5 because this paper looks better with the amendments but still has room for improvement.

---

> > > > > ### Author Response · Authors · 2022-12-07
> > > > > **Response to Reviewer 3P64**
> > > > >
> > > > > We sincerely thank you for your recognition of our hard work during the rebuttal process and for improving the scores.
> > > > > For the failures in the AntMaze tasks, we noticed that the reviewer Ax8y has a similar concern about it and thus we post a summary comment [Explanation for the AntMaze results](https://openreview.net/forum?id=o_HqtIc-oF&noteId=BhWREjsNwBZ), in which we provide a detailed explanation of the results in the AntMaze tasks.

---

### Official Review · Reviewer_AX8y · 2022-10-25

**Confidence:** 5
**Correctness:** 3
**Technical Novelty And Significance:** 2
**Empirical Novelty And Significance:** 2
**Recommendation:** 3

**Clarity, Quality, Novelty And Reproducibility:**

Regarding the presentation quality, the paper is well written and is easy to follow. The novelty of the proposed method is rather limited due to the fact that it is a small improvement on CQL.

**Strength And Weaknesses:**

I reviewed this paper for NeurIPS 2022 before it was withdrawn by the authors. Overall I think this paper presents an interesting direction towards improving value conservatism methods, where they often suffer from the problem of being overly conservative. However I do still have some concerns about the particular approach and empirical results.



### Strength

This paper presents a very interesting and potentially very effective direction for improving the use of value conservatism in Q-learning style offline RL algorithms. Conservative algorithms in offline RL are known to be difficult to tune, and often suffer the problem of either being overly conservative where the policy does not improve beyond the behavior policy, or being insufficiently conservative where the Q functions suffer from overestimation. Having adaptive conservatism coefficients that depend on the transition could be a promising way of solving this problem.



### Weaknesses

First of all, I want to emphasize (again) that certain baseline evaluations for CQL in this paper seem flawed, where a properly tuned CQL can achieve much better performance than reported in this paper. For example, in one open source CQL implementation I used, CQL can achieve 95 in hopper-medium-replay-v2 and the reported performance is 48, and in walker-medium-replay-v2, CQL can achieve 82 while the reported performance is only 26. Hence I believe that some of the CQL results are not valid, and this presents a major problem for this paper because the proposed algorithm directly improves upon CQL. I’ve brought up this point in my NeurIPS 2022 review of this paper and pointed out the open source CQL implementation suggested by the original CQL authors to reproduce these numbers. Unfortunately, the authors have not addressed this problem in this submission.


Besides the flaw in certain baseline evaluations, the empirical performance of the proposed method is also not very strong. Following the numbers on the D4RL suite, it seems that ACQL does not really outperform significantly, despite introducing many more degrees of freedom and complexity. It makes me question whether the marginal improvement is really due to the proposed method, or because of the extra tunable components introduced.

The specific choices of the loss functions to learn the conservative coefficients are not well justified. It is intuitive to see why the agent should be less conservative on good actions and more conservative on bad actions, but I’m not convinced that the proposed transition quality function is a good way of measuring this. Moreover, the overall loss formulation is quite complicated with many moving parts, so it is difficult to understand the necessity and choices of each part without good justifications.


**Summary Of The Paper:**

The paper focuses on the problem of offline reinforcement learning, where the authors consider the limitation of the conservative Q learning (CQL) method and propose a new way of applying a fine-grained control of the conservatism level. The authors argue that setting the level of conservatism to be a global constant as done in CQL is suboptimal, because the dataset often contains both good and poor actions and hence it is intuitive to put conservatism on good actions compared to the poor actions. Instead, the paper introduces adaptive coefficients to weight the conservatism terms.


Building on top of the formulation of CQL, the authors introduce two state-action dependent coefficients to weight the push up and push down loss instead of using a single global conservatism coefficient. The authors show theoretically that a properly chosen set of coefficients can still produce value estimates that lower bounds the true Q function, but with less conservatism than having a globally constant coefficient. The authors then propose a concrete formulation of the coefficient d based on the quality of the transition, which is measured by its relative reward and cumulative return magnitudes.

The authors evaluate the proposed method in the D4RL suite of tasks, and show that the proposed method improves upon the baseline CQL method in some environments.


**Summary Of The Review:**

Overall, while the general direction of research is interesting, due to the concerns about the empirical evaluations and justifications for the proposed method, I still cannot recommend acceptance of the paper in its current state.

---

> ### Author Response · Authors · 2022-11-17
> **Response to Reviewer AX8y**
>
> Thanks a lot for coordinating the review for our paper. We believe that based on the current code status and scoring system of different benchmark versions, we have done the best combination to produce fair results. It took us some time to set up additional experiments and please allow us to explain.
>
> (1) Regarding certain baseline evaluations for CQL in this paper.
>
> In Table 1, the results of DT, OnestepRL, and IQL are based on the mujuco v2 datasets and the rest results including CQL and ACQL are based on v0.
> And we directly report the official results of CQL from the D4RL whitepaper, so we think the result comparisons are fair.
> To say the least, even if CQL is flawed on the v0 datasets, that means ACQL is also flawed because we inherited its code.
> In fact, this comparison is unfair to ACQL because the results of other algorithms are not flawed.
> In addition, the results of DT, OnestepRL, and IQL on v2 dataset are usually better than the result on v0 because some bugs in v0 datasets are fixed in v2 datasets.
> That is also why we choose to report v0 results instead of v2, because we do not want to use v2 better results to compare with v0 results of other algorithms with bugs.
> As we report them directly from the D4RL whitepaper or their original papers, we also do not intentionally suppress the results of other algorithms.
>
> We really appreciate your suggestion at NIPs22, we actually tried to use the code you told us, but there were some bugs in the migration process that caused us to not use it in the end. After NIPs22, we mainly focused on improving the ACQL algorithm rather than running various code implementations and dataset versions and then reporting the best result.
>
> Now we have uploaded a new submission that notes all the versions of the datasets in all Tables.
> For Gym-MuJuCo environments, we provide two Tables (Table 1 in Section 6.1 and Table 7 in Appendix C.1) reporting the results on v0 and v2 respectively.
> We also add the tables to this comment as the following, please check the new results to see if there is any other question.

---

> > ### Author Response · Authors · 2022-11-17
> > **Response to Reviewer AX8y**
> >
> > (2) Regarding the empirical performance of ACQL is also not very strong.
> >
> > For Gym-MuJuCo environments, we provide two Tables (Table 1 in Section 6.1 and Table 7 in Appendix C.1) reporting the results on v0 and v2 respectively.
> > We believe these tables can help us see more clearly how ACQL compares to other algorithms.
> >
> > Table 1:  Normalized results on D4RL Gym-MuJoCo v0 environments.
> > |  Task Name  | BC | BCQ | BEAR | BRAC-v | BBRAC-p | AWR | TD3+BC | F-BRC | CQL | ACQL |
> > |  ----  | ----  | ----  | ----  | ----  | ----  | ----  | ----  | ----  | ----  | ----  |
> > | halfcheetah-e-v0 | 107.0 | - | 108.2 | -1.1 | 3.8 | - | 105.7 | **108.4** | 103.5 | 97.7 |
> > | halfcheetah-m-e-v0  | 35.8 | 64.7 | 53.4 | 41.9 | 44.2 | 52.7 | **97.9** | 93.3 | 62.4 | 93.2 |
> > | halfcheetah-m-r-v0 | 38.4 | 38.2 | 38.6 | **47.7** | 45.4 | 40.3 | 43.3 | 43.2 | 46.2 | 46.4 |
> > | halfcheetah-m-v0 | 36.1 | 40.7 | 41.7 | **46.3** | 43.8 | 37.4 | 42.8 | 41.3 | 44.4 | 42.8 |
> > | halfcheetah-r-v0 | 2.1 | 2.2 | 25.1 | 31.2 | 24.1 | 2.5 | 10.2 | 33.3 | **35.4** | 31.3 |
> > | halfcheetah-sum | 219.4 | - | 267.0 | 166.0 | 161.3 | - | 299.9 | **319.5** | 291.9 | 311.4 |
> > |  ----  | ----  | ----  | ----  | ----  | ----  | ----  | ----  | ----  | ----  | ----  |
> > | hopper-e-v0 | 109.0 | - | 110.3 | 3.7 | 6.6 | - | 112.2 | **112.3** | 112.2 | **116.3** |
> > | hopper-m-e-v0 | 111.9 | 110.9 | 96.3 | 0.8 | 1.9 | 27.1 | 112.2 | **112.4** | 98.7 | 111.8 |
> > | hopper-m-r-v0 | 11.8 | 33.1 | 33.7 | 0.6 | 0.6 | 28.4 | 31.4 | 35.6 | **48.6** | **77.3** |
> > | hopper-m-v0 | 29.0 | 54.5 | 52.1 | 31.1 | 32.7 | 35.9 | **99.5** | 99.4 | 58.0 | 85.8 |
> > | hopper-r-v0 | 9.8 | 10.6 | 11.4 | **12.2** | 11.0 | 10.2 | 11.0 | 11.3 | 10.8 | **12.7** |
> > | hopper-sum | 271.5 | - | 303.8 | 48.4 | 52.8 | - | 366.3 | **371.0** | 328.3 | **403.9** |
> > |  ----  | ----  | ----  | ----  | ----  | ----  | ----  | ----  | ----  | ----  | ----  |
> > | walker-e-v0 | **125.7** | - | 106.1 | 0.0 | -0.2 | - | 105.7 | 103.0 | 107.2 | 116.1 |
> > | walker-m-e-v0 | 6.4 | 57.5 | 40.1 | 81.6 | 76.9 | 53.8 | 101.1 | 105.2 | **111.0** | 94.5 |
> > | walker-m-r-v0 | 11.3 | 15.0 | 19.2 | 0.9 | -0.3 | 15.5 | 25.2 | **41.8** | 26.7 | **45.2** |
> > | walker-m-v0 | 6.6 | 53.1 | 59.1 | **81.1** | 77.5 | 17.4 | 79.7 | 78.8 | 79.2 | **81.8** |
> > | walker-r-v0 | 1.6 | 4.9 | **7.3** | 1.9 | -0.2 | 1.5 | 1.4 | 1.5 | 7.0 | **11.8** |
> > | walker-sum | 151.6 | - | 231.8 | 165.6 | 153.7 | - | 313.1 | 330.3 | **331.1** | **349.4** |
> >
> > Table 7:  Normalized results on D4RL Gym-MuJoCo v2 environments.
> > |  Task Name  | BC | 10%BC | DT | AWAC | OnestepRL | TD3+BC | IQL | CQL | ACQL |
> > |  ----  | ----  | ----  | ----  | ----  | ----  | ----  | ----  | ----  | ----  |
> > | halfcheetah-m-e-v2 | 55.2 | 92.9 | 86.8 | 36.8 | **93.4** | 90.7 | 86.7 | 91.6 | 87.9 |
> > | halfcheetah-m-r-v2 | 36.6 | 40.6 | 36.6 | 40.5 | 38.1 | 44.6 | 44.2 | 45.5 | **46.3** |
> > | halfcheetah-m-v2 | 42.6 | 42.5 | 42.6 | 37.4 | **48.4** | 48.3 | 47.4 | 44.0 | **48.4** |
> > | halfcheetah-r-v2 | 2.3 | 2.0 | 2.2 | 2.2 | 6.9 | 11.0 | - | 18.6 | **25.9** |
> > | halfcheetah-sum | 136.7 | 178.0 | 168.2 | 116.9 | 186.8 | 194.6 | 178.3 | 199.7 | **208.5** |
> > |  ----  | ----  | ----  | ----  | ----  | ----  | ----  | ----  | ----  | ----  | ----  |
> > | hopper-m-e-v2 | 52.5 | **110.9** | 107.6 | 80.9 | 103.3 | 98.0 | 91.5 | 105.4 | 82.2 |
> > | hopper-m-r-v2 | 18.1 | 75.9 | 82.7 | 37.2 | **97.5** | 60.9 | 94.7 | 95.0 | **97.9** |
> > | hopper-m-v2 | 52.9 | 56.9 | 67.6 | **72.0** | 59.6 | 59.3 | 66.3 | 58.5 | **91.4** |
> > | hopper-r-v2 | 4.8 | 4.1 | 7.5 | **9.6** | 7.8 | 8.5 | - | 9.3 | **31.4** |
> > | hopper-sum | 128.3 | 247.8 | 265.4 | 199.7 | 268.2 | 226.7 | 252.5 | 268.2 | **302.9** |
> > |  ----  | ----  | ----  | ----  | ----  | ----  | ----  | ----  | ----  | ----  | ----  |
> > | walker-m-e-v2 | 107.5 | 109.0 | 108.1 | 42.7 | **113.0** | 110.1 | 109.6 | 108.8 | 108.3 |
> > | walker-m-r-v2 | 26.0 | 62.5 | 66.6 | 27.0 | 49.5 | **81.8** | 73.9 | 77.2 | **85.5** |
> > | walker-m-v2 | 75.3 | 75.0 | 74.0 | 30.1 | 81.8 | **83.7** | 78.3 | 72.5 | **84.9** |
> > | walker-r-v2 | 1.7 | 1.7 | 2.0 | 5.1 | **6.1** | 1.6 | - | 2.5 | 5.2 |
> > | walker-sum | 210.5 | 248.2 | 250.7 | 104.9 | 250.4 | 277.2 | 261.8 | 261.0 | **283.9** |
> >
> > From these tables, we can observe that ACQL consistently outperformed other baselines on both v0 and v2 dataset of all 3 kinds of environments, especially when the dataset contains more medium and random data.
> > From the perspective of the dataset types, ACQL is very a balanced algorithm that achieves excelling results on all kinds of datasets with expert, medium and random data.
> > For instance, in Table 1, TD3+BC delivered relatively higher returns on datasets with better quality such as 105.7 on walker-expert while lower returns on random data like 1.4 on walker-random.
> > The CQL has a similar problem in that it is difficult for CQL to achieve satisfactory performance on all kinds of dataset with a fixed $\alpha$.

---

> > > ### Author Response · Authors · 2022-11-17
> > > **Response to Reviewer AX8y**
> > >
> > > (3) Regarding whether the marginal improvement is really due to the proposed method, or because of the extra tunable components introduced.
> > >
> > > After NIPs22, We have carefully considered your comments and focused on how to improve ACQL.
> > > One issue in the submission of NIPs22 is that there are many components and hyperparameters that need to be tuned, like the $C_1, C_2$.
> > > In this submission, we have significantly reduced the number of hyperparameters that need to be tuned.
> > > Now we only need to set the $\alpha$ like what we do in CQL and other hyperparameters can be calculated automatically according to the given dataset.
> > > To achieve this, firstly we define relative transition quality $m(s,a)$ considering both the single-step reward and whole discounted trajectory MC return.
> > > The relative transition quality $m(s,a)$ is calculated automatically with different datasets and its range is (0, 1).
> > > So we can easily shift and scale its value to match the magnitude of any variables suitable for different datasets.
> > > Based on the relative transition quality $m(s,a)$, we also calculate the values of $C_1, C_2$ automatically with different datasets.
> > > There are no other extra hyperparameters that need to be tuned and we list all the hyperparameters in Table 6 in Appendix B.
> > >
> > > Compared to CQL, ACQL is actually simpler to tune, because we can set a relatively large $\alpha$ value (e.g., 20) for ACQL, and then ACQL will limit the Q-values between the real Q-values and the CQL Q-values, and the exact values will be calculated adaptively.
> > > In contrast, we need to adjust different $\alpha$ values for CQL in order to get satisfactory results on different datasets (e.g., 1 for random datasets and 20 for expert datasets).
> > > The fact that we conduct the experiments on 15 mujuco v2 datasets and achieve state-of-art results in less than two weeks also demonstrates that ACQL does not need to tune lots of hypermeters.
> > >
> > > (4) Regarding the proposed transition quality function is not well justified.
> > >
> > > As we stated in Definition 4.1 in Section 4, the best solution is the optimal Q-function $Q^*$.
> > > However, it is an ill-posed problem since if we know the optimal Q-function, we can directly train an optimal policy
> > > and solve the problem.
> > > Since we do not know the optimal transition quality, it is too difficult to directly compare the proposed transition quality function and the optimal transition quality.
> > > We believe that the results of ACQL on so many different domains and datasets demonstrate that the proposed transition quality function can work well.
> > > Of course, the optimal transition quality may not be the best one, and we also did not claim it is the best solution.
> > > As we discussed in Section 5.1, we think it does have some room for improvement, for example, in combination with other methods including the "upper envelope" in BAIL[1] and uncertainty estimation in [2] has the potential to be incorporated in ACQL.
> > > In this work, we mainly focus on the ACQL framework that can control the conservative level more adaptively and thus does not make the algorithm too complex.
> > > How to combinate ACQL with other methods are out of the scope of this paper and we left it to future work.
> > >
> > > (5) Regarding the overall loss formulation is quite complicated.
> > >
> > > For the effect of the proposed losses, we have built extensive ablation studies in Section 6.3 and in Appendix C.2 (old submission), and Appendix C.4 (new submission).
> > > From the results of these ablation studies, we can see each loss contributes to the final results.
> > > Although the loss formulation is formally very complex, with 4 different loss items, they do not clash as strongly as traditional multi-task learning in the actual optimization process, because these losses are only constraining conditions instead of real targets.
> > > Furthermore, we relax the conditions and use $C_1, C_2$ as the soft margins of the conservative level compared to the true Q-function and CQL respectively, thus also reducing the conflicts among these losses.
> > >
> > > [1] Chen, Xinyue, et al. "BAIL: Best-action imitation learning for batch deep reinforcement learning." Advances in Neural Information Processing Systems 33 (2020): 18353-18363.
> > > [2] Yu, Tianhe, et al. "Mopo: Model-based offline policy optimization." Advances in Neural Information Processing Systems 33 (2020): 14129-14142.

---

> > > > ### Comment · Reviewer_AX8y · 2022-11-18
> > > > **Re: Response to Reviewer AX8y**
> > > >
> > > > I'd like the thank the authors for the detailed response and additional experiment results.
> > > >
> > > > ### D4RL Gym v0 vs v2 Tasks
> > > > First of all I want to point out that v0 tasks in D4RL should no longer be used to evaluate algorithms. The data collection for the v0 tasks has some bugs which were fixed in the v2 dataset. Therefore, I believe that v0 tasks should not be used since getting better performance on the v0 tasks is likely a result of overfitting to the flawed data collection process. In the new results of the v2 tasks that the authors provide here, it seems that the ACQL algorithm does not outperform CQL significantly. Since some of these numbers are slight worse than a well-tuned CQL baseline, I believe that the improvement range here fall within the variation of hyperparameter tuning of CQL, and hence does not demonstrate the benefit of ACQL reliably.
> > > >
> > > > ### Improvement Due to Tunable Components
> > > > Regarding the tunable components, I'm not suggesting that the proposed method introduces hyperparameters in the traditional sense. What I'm concerned about is that the proposed rules to calculate these quantities automatically from data are introducing tunable components implicitly, especially when the rules used in this paper are rather complicated. Following the intuition proposed by the authors, there are many alternative rules that one could choose to achieve the same goal, and it is very likely that most of these alternatives don't work as well as the proposed method. Therefore, selecting the rule that works the best is essentially the same as hyperparameter tuning, which is susceptible to overfitting to a particular set of tasks.
> > > >
> > > > ### Complexity and Justification
> > > > Following my previous point, given the complexity of the proposed method and the limited empirical improvement, an effective way of showing that the proposed method works not because of the extra tunable components is through theoretical analysis. If  the authors can show that the rules used to calculate C1, C2, transition quality and the adaptive weight loss is justifiable through theoretical derivations, then it is possible to believe that the overall complexity of the proposed method is not merely a result of overfitting. Unfortunately, the authors have not provided such analysis in the paper.
> > > >
> > > >
> > > > Overall I will keep my evaluation of the paper.

---

> > > > > ### Author Response · Authors · 2022-11-18
> > > > > **Response to Reviewer AX8y**
> > > > >
> > > > > (1) Regarding D4RL Gym v0 vs v2 Tasks
> > > > >
> > > > > 1.1 Regarding "v0 tasks in D4RL should no longer be used to evaluate algorithms".
> > > > >
> > > > > Firstly, not all v0 datsets have bugs.
> > > > > According to our knowledge, the issue about the bugs is here [Error in hopper replay datasets](https://github.com/Farama-Foundation/D4RL/issues/86) and there are only hopper datasets that have some bugs.
> > > > > So the results on other v0 datasets are still valid.
> > > > >
> > > > > Secondly, the official statement about the bugs in v0 datasets is "As well as some bugfixes: All trajectories now timeout at 1000 steps." in the official repository [D4RL repo](https://github.com/Farama-Foundation/d4rl/wiki/Tasks).
> > > > > The bugs are only about the trajectory's length, so they will not affect the final performance too much.
> > > > >
> > > > > Thirdly, many state-of-art methods like [1,2,3] and all results from the D4RL whitepaper[4] are evaluated on the v0 datasets, are you suggesting that all their results are also invalid?
> > > > > We believe that even if there are small flaws in individual data in v0 tasks, the overall results are still worthy of reference.
> > > > >
> > > > > 1.2 Regarding "some of these numbers are slight worse than a well-tuned CQL baseline"
> > > > >
> > > > > For the v2 tasks, all the results of baselines are taken from the survey[5], IQL[6], and their original papers.
> > > > > As we report them directly from another place, we **do not and can not** suppress the results of other baseline algorithms.
> > > > >
> > > > > 1.3 Regarding "ACQL algorithm does not outperform CQL significantly" and "improvement range here fall within the variation of hyperparameter tuning of CQL"
> > > > >
> > > > > Firstly, in Table 7 in Appendix C.1, we have shown ACQL can outperform CQL for most of the v2 tasks.
> > > > > For example, in hopper-m-v2, ACQL can achieve **91.4** while CQL only achieves **58.5**.
> > > > > And in hopper-r-v2, ACQL can achieve **31.4** while CQL only achieves **9.3**, which is more than a 300% improvement.
> > > > >
> > > > > Secondly, for the v2 tasks, we do not tune hyperparameters and directly **set $\alpha=20$ for all 15 tasks** and its final performance are so good.
> > > > > So we believe that ACQL has the potential to achieve better performance once we tune the hyperparameters carefully.
> > > > > In contrast, CQL has to tune its hyperparameters especially the $\alpha$ to achieve satisfactory performance on different datasets (e.g., large $\alpha$ for expert datasets and small $\alpha$ for random datasets).
> > > > >
> > > > > Thirdly, the results on v2 datasets of CQL are from the IQL[6] and we believe it is clearly a fair comparison.

---

> > > > > > ### Comment · Reviewer_AX8y · 2022-11-28
> > > > > > **Response to Authors**
> > > > > >
> > > > > > I'd like to thank the authors for the detailed response. Here are my comments.
> > > > > >
> > > > > > > Firstly, not all v0 datsets have bugs. According to our knowledge, the issue about the bugs is here Error in hopper replay datasets and there are only hopper datasets that have some bugs. So the results on other v0 datasets are still valid.
> > > > > >
> > > > > > According to the original authors of D4RL, besides the bugs in the hopper datasets, all the Gym datasets have some portion of terminal flags marked incorrectly. This would result in truncated backup for Q learning style algorithms, so they should not be used for benchmarking algorithms anymore.
> > > > > >
> > > > > >
> > > > > >  > 3.1 Regarding "an effective way of showing that the proposed method works not because of the extra tunable components is through theoretical analysis"
> > > > > >
> > > > > > I want to clarify that not all methods need to have theoretical analysis to support them. However, in this specific case, the proposed method here is quite complicated with a lot of unintuitive rules added on top of CQL, and the performance is not significantly better than CQL. Therefore, a very likely alternative hypothesis is that the proposed method works better simply because of these extra tunable components. Hence, I think there are three ways to justify the proposed method: (1) greatly simplify the set of rules to make them the intuitive choice; (2) leave these rules as is, but provide theoretical explanations; (3) show that these rules provide significant performance improvement compared to CQL, beyond what's possible for the added tunable parts. I'd be happy to see either one of these, but unfortunately the paper authors have not yet provided any of such justifications.
> > > > > >
> > > > > >
> > > > > > > For the v2 tasks, all the results of baselines are taken from the survey[5], IQL[6], and their original papers. As we report them directly from another place, we do not and can not suppress the results of other baseline algorithms.
> > > > > >
> > > > > > The fact that the performance improvement over CQL lies within the range of implementation variations further supports the alternative hypothesis that the proposed method works better simply because of these extra tunable components.
> > > > > >
> > > > > >
> > > > > > > Secondly, for the v2 tasks, we do not tune hyperparameters and directly set $\alpha = 20$ for all 15 tasks and its final performance are so good.
> > > > > >
> > > > > > Please note that the CQL numbers are also obtained by using a single alpha across all v2 tasks.
> > > > > >
> > > > > >
> > > > > > Finally, given that the proposed method also works significantly worse than CQL on the more difficult antmaze environments, I'd like to keep my original evaluation of the paper.

---

> > > > > > > ### Author Response · Authors · 2022-12-03
> > > > > > > **Response to Reviewer AX8y**
> > > > > > >
> > > > > > > We sincerely thank you valuable feedback and respect your insistence.
> > > > > > >
> > > > > > > (1) Regarding D4RL Gym v0 vs v2 Tasks.
> > > > > > >
> > > > > > > > According to the original authors of D4RL, besides the bugs in the hopper datasets, all the Gym datasets have some portion of terminal flags marked incorrectly. This would result in truncated backup for Q learning style algorithms, so they should not be used for benchmarking algorithms anymore.
> > > > > > >
> > > > > > > We totally agree with your opinion. Since we have built the experiments on mujuco v2 tasks, we think this issue has been solved.
> > > > > > > We will move the mujuco-v2 results into our main manuscript and move the mujuco-v0 results into the appendix for any reader who wants to check more details.
> > > > > > > Do you agree with that?
> > > > > > >
> > > > > > >
> > > > > > > (2) Regarding the complicated proposed method with a lot of unintuitive rules added on top of CQL.
> > > > > > >
> > > > > > > Due to the page limitation, we really did not have more space to elaborate on the idea behind the ACQL implementation (i.e., from Equations 8 to 13).
> > > > > > > As the designers of the algorithm, since we are already familiar with it, we naturally felt that it is not very complicated.
> > > > > > > We think it is not the only way to instantiate ACQL, and it has the potential to get more improvement by combining with other methods as mentioned in Section 5.1.
> > > > > > > This is also why we pay more attention to describing the motivation and properties of the ACQL framework rather than the execution part of ACQL.
> > > > > > > We apologize that we have made you confused by writing only from our own perspective.
> > > > > > >
> > > > > > > Recall that CQL is also an instance of the ACQL framework with the simplest rules (i.e., set the adaptive weights directly using a fixed $\alpha$).
> > > > > > > And we believe that CQL can work well and also illustrates the effectiveness of the ACQL framework to some extent.
> > > > > > >
> > > > > > > Overall, we believe it is only a misunderstanding and thus we post a summary comment [Explanation for the design of ACQL](https://openreview.net/forum?id=o_HqtIc-oF&noteId=KzjyyznnVj), in which we provide a detailed explanation about the idea behind the complicated rules and conduct additional experiments with different simplified rules to justify the proposed method.
> > > > > > > Please do not hesitate to ask if you have more questions or concerns.
> > > > > > > We are willing to provide more clarifications or experiments in these regards.

---

> > > > > > > > ### Author Response · Authors · 2022-12-03
> > > > > > > > **Response to Reviewer AX8y Cont'd**
> > > > > > > >
> > > > > > > > (3) Regarding the performance of ACQL is not significantly better than CQL and the CQL numbers are also obtained by using a single alpha across all v2 tasks.
> > > > > > > >
> > > > > > > > As mentioned in the third paragraph in Introduction Section, there is usually a huge performance gap between CQL with different $\alpha$ values on different types of datasets (e.g., a larger $\alpha$ value usually can achieve better performance on expert datasets).
> > > > > > > > Thus one main goal is to design a balanced algorithm that can calculate the weight adaptively according to the datasets without any prior about the dataset (i.e., we do not know if the dataset is an expert one or a random one).
> > > > > > > >
> > > > > > > > Considering that CQL needs well-tuned and different $\alpha$ values to achieve the best performance on each type of dataset, it is unfair to require ACQL to beat all the best CQL results at the same time.
> > > > > > > > In practice, we believe that you can also feel that it is difficult to tune the $\alpha$ value of CQL for each kind of dataset.
> > > > > > > >
> > > > > > > > > The CQL numbers are also obtained by using a single alpha across all v2 tasks.
> > > > > > > >
> > > > > > > > We carefully checked the IQL paper[1] where the CQL results are taken from and did not find any statement indicating that all CQL numbers are obtained by using a single $\alpha$ value.
> > > > > > > > To further evaluate the CQL performance with a single $\alpha$ value, following your suggestion at NIPs22 and IQL paper, we used the bug-free code from this GitHub repo [CQL implementations](https://github.com/young-geng/CQL) to conduct comprehensive experiments for CQL with different $\alpha$ values from {1, 5, 10, 20}.
> > > > > > > > All results are average values of 3 random seeds.
> > > > > > > >
> > > > > > > > As shown in the following Table, we can see your statement is right and the CQL implementation in IQL is likely to use $\alpha = 5$, which is a compromised value for all kinds of datasets.
> > > > > > > >
> > > > > > > > Even so, ACQL outperformed CQL-5 in terms of the sum results on all three environments.
> > > > > > > > We can observe that ACQL usually achieved better results on medium and random datasets.
> > > > > > > > It is because the transitions with different qualities are mixed up, and ACQL can adaptively calculate the weights according to their relative qualities.
> > > > > > > >
> > > > > > > > In hopper-m-v2, ACQL can achieve 91.4 while the best one CQL-1 achieves 74.8.
> > > > > > > > And in hopper-r-v2, ACQL can achieve 31.4 while the best one CQL-20 only achieves 9.3.
> > > > > > > > If CQL uses an unsuitable $\alpha$ value (e.g., CQL-1 on halfcheetah-m-e-v2), the performance gap can be much bigger.
> > > > > > > > Thus we argue that performance improvements of ACQL over CQL do not lie within the range of implementation variations.
> > > > > > > >
> > > > > > > >
> > > > > > > > Table: Comparison of different conservative levels of CQL on D4RL Gym-MuJoCo v2 environments.
> > > > > > > > |  Task Name  | CQL-1 | CQL-5 | CQL-10 | CQL-20 | CQL[1,2] | ACQL-20 |
> > > > > > > > |  :----:  | :----:  | :----:  | :----:  | :----:  | :----:  | :----: |
> > > > > > > > | halfcheetah-m-e-v2 | 55.6 | 77.4 | 67.3 | 84.2 | **91.6** | 87.9 |
> > > > > > > > | halfcheetah-m-r-v2 | 14.5 | **47.6** | 45.3 | 43.9 | 45.5 | 46.3 |
> > > > > > > > | halfcheetah-m-v2 | **54.9** | 48.6 | 46.7 | 45.1 | 44.0 | 48.4 |
> > > > > > > > | halfcheetah-r-v2 | **29.2** | 25.4 | 20.8 | 6.6 | 18.6 | 25.9 |
> > > > > > > > | halfcheetah-sum | 154.2 | 199.0 | 180.1 | 179.8 | 199.7 | **208.5** |
> > > > > > > > |  ----  | ----  | ----  | ----  | ----  | ----  | ----  | ----  | ---- |
> > > > > > > > | hopper-m-e-v2 | 67.6 | 106.9 | **111.6** | 83.4 | 105.4 | 82.2 |
> > > > > > > > | hopper-m-r-v2 | **101.4** | 95.7 | 98.8 | 65.1 | 95.0 | 97.9 |
> > > > > > > > | hopper-m-v2 | 74.8 | 60.9 | 60.8 | 71.4 | 58.5 | **91.4** |
> > > > > > > > | hopper-r-v2 | 5.5 | 4.6 | 8.3 | 8.4 | 9.3 | **31.4** |
> > > > > > > > | hopper-sum | 249.3 | 268.1 | 279.5 | 228.33 | 268.2 | **302.9** |
> > > > > > > > |  ----  | ----  | ----  | ----  | ----  | ----  | ----  | ----  | ---- |
> > > > > > > > | walker-m-e-v2 | 106.6 | **109.9** | 109.6 | 108.5 | 108.8 | 108.3 |
> > > > > > > > | walker-m-r-v2 | 32.1 | 80.2 | 79.3 | 63.9 | 77.2 | **85.5** |
> > > > > > > > | walker-m-v2 | 49.2 | 83.9 | 81.3 | 81.7 | 72.5 | **84.9** |
> > > > > > > > | walker-r-v2 | 2.5 | 2.9 | **8.3** | 5.1 | 2.5 | 5.2 |
> > > > > > > > | walker-sum | 190.4 | 276.9 | 278.5 | 259.2 | 261.0 | **283.9** |

---

> > > > > > > > > ### Author Response · Authors · 2022-12-03
> > > > > > > > > **Response to Reviewer AX8y Cont'd**
> > > > > > > > >
> > > > > > > > > (4) Regarding the results of AntMaze tasks.
> > > > > > > > >
> > > > > > > > > > Finally, given that the proposed method also works significantly worse than CQL on the more difficult antmaze environments, I'd like to keep my original evaluation of the paper.
> > > > > > > > >
> > > > > > > > > We argue that one primary reason is that the task on the antmaze environments is a sparse reward problem and rewards are 0 or 1.
> > > > > > > > > In that case, it is too difficult to calculate the adequate transition quality using Equations 8 and 9, leading to task failure, which is not unexpected for us.
> > > > > > > > > Recall that our main goal is to design a balanced algorithm for datasets with different data qualities (e.g., expert, medium, and random datasets), it is out of scope to require ACQL to solve the sparse problem.
> > > > > > > > >
> > > > > > > > > On the other hand, note that CQL is a specified case of ACQL framework (i.e., set weights using a fixed $\alpha$), its success on AntMaze tasks shows that the compatibility of ACQL framework, under which everyone can design their own adaptive weight functions for different kinds of scenarios.
> > > > > > > > > We never claim the current implementation is the only and best solution and we would welcome any continual improvement under the umbrella of ACQL.
> > > > > > > > >
> > > > > > > > > [1] Prudencio, Rafael Figueiredo, Marcos ROA Maximo, and Esther Luna Colombini. "A Survey on Offline Reinforcement Learning: Taxonomy, Review, and Open Problems." arXiv preprint arXiv:2203.01387 (2022).
> > > > > > > > >
> > > > > > > > > [2] Kostrikov, Ilya, et al. "Offline reinforcement learning with fisher divergence critic regularization." International Conference on Machine Learning. PMLR, 2021.

---

> > > > > ### Author Response · Authors · 2022-11-18
> > > > > **Response to Reviewer AX8y**
> > > > >
> > > > > (2) Regarding Improvement Due to Tunable Components
> > > > >
> > > > >
> > > > > 2.1 Regarding "proposed rules to calculate these quantities automatically from data are introducing tunable components implicitly, especially when the rules used in this paper are rather complicated".
> > > > >
> > > > > Firstly, when the $Q^*$ cannot be computed and the normal Q-function is not reliable in the offline setting, the rest choice to measure the quality of data is the single-step reward and MC return of each transition.
> > > > > Thus we both consider the single-step reward and MC return to calculate the relative transition quality $m(s,a)$.
> > > > > Equations 8 and 9 just look complicated, but the goal is actually very simple and is to get its range from 0 to 1.
> > > > > So we can easily shift and scale its value to match the magnitude of any variables suitable for different datasets.
> > > > >
> > > > > Secondly, there are many complicated offline DRL methods that also introduce tunable components that can perform very well on these tasks, like all the model-based offline DRL methods[7,8].
> > > > > So whether an algorithm is complex can not be used as a criterion to judge whether an algorithm is good or bad.
> > > > >
> > > > > Thirdly, one of the main contributions of this paper is that we propose ACQL **framework**, in which we aim to keep the conservative level in an adaptive range and the adaptive weight function in this range should have monotonicity.
> > > > > Even if the rule for calculating the quantities seems very complex to you, it is only a minor detail at the implementation level and does influence the whole paper.
> > > > >
> > > > > 2.2 Regarding "there are many alternative rules that one could choose to achieve the same goal, and it is very likely that most of these alternatives don't work as well as the proposed method."
> > > > >
> > > > > Firstly, one of the main contributions of this paper is that we propose ACQL **framework**, which sheds the light on how to design a proper conservative Q-function.
> > > > > In the ACQL framework, other users can design their own adaptive weight functions as they desire.
> > > > > For example, if they have prior knowledge about the dataset, they can change the rules.
> > > > > We also do not claim the current implementation is the best solution and we would welcome any continual improvement under the umbrella of ACQL.
> > > > >
> > > > > Secondly, the statement here is your guess and it is impossible to evaluate since there are infinite kinds of adaptive weight function designs under the ACQL framework.
> > > > > There must be that some can work well and some fail.
> > > > > In this paper, we have proposed one solution which can perform very well on many domains and datasets.
> > > > > So why focus on what is not presented in the paper?
> > > > >
> > > > > 2.3 Regarding "selecting the rule that works the best is essentially the same as hyperparameter tuning, which is susceptible to overfitting to a particular set of tasks."
> > > > >
> > > > > Firstly, according to your statement here, every algorithm which works well can be regarded as a kind of hyperparameter tuning explicitly or implicitly to overfit a particular set of tasks. It makes no sense to question the need to tune hyperparameters.
> > > > >
> > > > > Secondly, in this paper, we have proposed one solution which can perform very well on many domains and datasets.
> > > > > If the users do not want to design their own adaptive weight functions, they at least can follow our implementation.
> > > > >
> > > > > Thirdly, in ACQL, we have conducted extensive experiments on Gym-MuJoCo v0 v2 Tasks, Adroit Tasks, Franka Kitchen Tasks, AntMaze Tasks, and for each task there are several different types of datesets.
> > > > > We use the same rule for all these experiments and believe these results have demonstrated ACQL's effectiveness and wide range of domains.

---

> > > > > ### Author Response · Authors · 2022-11-18
> > > > > **Response to Reviewer AX8y**
> > > > >
> > > > > (3) Regarding Complexity and Justification
> > > > >
> > > > >
> > > > > 3.1 Regarding "an effective way of showing that the proposed method works not because of the extra tunable components is through theoretical analysis"
> > > > >
> > > > > Firstly, of course, theoretical analysis is an effective way to support the proposed method.
> > > > > However, it is not the only way.
> > > > > According to your statement here, most deep learning methods (e.g., ResNet[9]) are not justifiable but they still work very well in our daily life.
> > > > > In ACQL, we have conducted extensive experiments on Gym-MuJoCo v0 v2 Tasks, Adroit Tasks, Franka Kitchen Tasks, and AntMaze Tasks and for each task, there are several different types of datasets.
> > > > > We use the same rule for all these experiments and believe these results have demonstrated ACQL's effectiveness in a wide range of domains.
> > > > >
> > > > > Secondly, we provided the theoretical analysis and the corresponding proofs of the ACQL framework in Section 4 and Appendix A.
> > > > > In the ACQL framework, we claimed that once the proposed constraints and conditions are satisfied, ACQL can learn a conservative Q-function.
> > > > > In our implementation, the rules used to calculate C1, C2, transition quality, and adaptive weight loss are used to satisfy the constraints, thus there is no specified theoretical analysis for the implementation.
> > > > > Your requirement about the theoretical analysis of the detailed implementation is similar to that asking the theoretical analysis of why using a 3-layer network instead of a 4-layer network, which is unnecessary and unrealistic.
> > > > >
> > > > > Thirdly, there exists a lot of state-of-art methods[1,10,11] which do not have theoretical analysis and we think the theoretical analysis is an aid to help understand the properties of algorithms and is not a necessary part.
> > > > >
> > > > > Fourthly, there exists a huge gap between the theoretical results and the empirical results.
> > > > > For example, BCQ [12] provided the theoretical analysis that it can converge to the optimal value function under the dataset MDP.
> > > > > However, many new state-of-art methods take it as the baseline method and can beat it.
> > > > >
> > > > > Fifthly, we do not mainly focus on RL theory and our paper is also not a purely theoretical study of the RL theory.
> > > > > We want to emphasize that the extensive experiments on Gym-MuJoCo v0 v2 tasks, Adroit tasks, Franka Kitchen tasks, and AntMaze tasks have demonstrated ACQL's effectiveness in a wide range of domains.
> > > > > Please do not ignore the main contribution of this paper and focus on the specified details.
> > > > >
> > > > > [1] Fujimoto, Scott, and Shixiang Shane Gu. "A minimalist approach to offline reinforcement learning." Advances in neural information processing systems 34 (2021): 20132-20145.
> > > > >
> > > > > [2] Kostrikov, Ilya, et al. "Offline reinforcement learning with fisher divergence critic regularization." International Conference on Machine Learning. PMLR, 2021.
> > > > >
> > > > > [3] Kumar, Aviral, et al. "Conservative q-learning for offline reinforcement learning." Advances in Neural Information Processing Systems 33 (2020): 1179-1191.
> > > > >
> > > > > [4] Fu, Justin, et al. "D4rl: Datasets for deep data-driven reinforcement learning." arXiv preprint arXiv:2004.07219 (2020).
> > > > >
> > > > > [5] Prudencio, Rafael Figueiredo, Marcos ROA Maximo, and Esther Luna Colombini. "A Survey on Offline Reinforcement Learning: Taxonomy, Review, and Open Problems." arXiv preprint arXiv:2203.01387 (2022).
> > > > >
> > > > > [6] Kostrikov, Ilya, et al. "Offline reinforcement learning with fisher divergence critic regularization." International Conference on Machine Learning. PMLR, 2021.
> > > > >
> > > > > [7] Matsushima, Tatsuya, et al. "Deployment-efficient reinforcement learning via model-based offline optimization." arXiv preprint arXiv:2006.03647 (2020).
> > > > >
> > > > > [8] Yu, Tianhe, et al. "Combo: Conservative offline model-based policy optimization." Advances in neural information processing systems 34 (2021): 28954-28967.
> > > > >
> > > > > [9] He, Kaiming, et al. "Deep residual learning for image recognition." Proceedings of the IEEE conference on computer vision and pattern recognition. 2016.
> > > > >
> > > > > [10] Chen, Lili, et al. "Decision transformer: Reinforcement learning via sequence modeling." Advances in neural information processing systems 34 (2021): 15084-15097.
> > > > >
> > > > > [11] Brandfonbrener, David, et al. "Offline rl without off-policy evaluation." Advances in Neural Information Processing Systems 34 (2021): 4933-4946.
> > > > >
> > > > > [12] Fujimoto, Scott, David Meger, and Doina Precup. "Off-policy deep reinforcement learning without exploration." International conference on machine learning. PMLR, 2019.

---

### Author Response · Authors · 2022-12-03
**Response to Reviewers AX8y, 3P64, KtHP**

We thank all the reviewers for the time and expertise they have invested in these reviews and for their positive and constructive feedback.
Your comments and suggestions have helped us to improve the paper.
We provide a response and clarifications below for the common concerns and hope they can address your concerns.
Since we cannot upload a newer version of the paper now, we promise we will add all experimental results in this rebuttal and more explanation about the ACQL implementation into our final version of the paper.

(1) **Regarding D4RL Gym v0 vs v2 Tasks.**

For Gym-MuJuCo environments, we provide two Tables (Table 1 in Section 6.1 and Table 7 in Appendix C.1) reporting the results on v0 and v2 respectively. We believe these tables can help you see more clearly how ACQL compares to other algorithms.
We also added the tables to this comment [Gym-MuJuCo v0 and v2 results](https://openreview.net/forum?id=o_HqtIc-oF&noteId=FNkyRJzapi6).

Considering the v0 datasets have some internal bugs and are out-of-date, we will move the mujuco-v2 results into our main manuscript and move the mujuco-v0 results into the appendix for any reader who wants to check more details.

(2) **Regarding the complicated rules for ACQL implementation (i.e., Equations 8-13).**

Due to the page limitation, we really did not have more space to elaborate on the idea behind the ACQL implementation (i.e., from Equations 8 to 13).
As the designers of the algorithm, since we are already familiar with it, we naturally felt that it is not very complicated.
Before we give the explanation, we want to emphasize that it is not the only way to instantiate ACQL, and it has the potential to get more improvement by combining with other methods as mentioned in Section 5.1.
This is also why we pay more attention to describing the motivation and properties of the ACQL framework rather than the execution part of ACQL.
We apologize that we have made you confused by writing only from our own perspective.

**Explanation for the design of ACQL implementation**

Following the idea from Section 4, we need to set higher conservative levels for "bad" actions and lower conservative levels for "good" actions.
The major question at hand is "how do we judge a good or bad action?".

2.1 **Explanation for Equation 8**

In online RL setting, we usually train a Q-function to predict these values.
In offline RL setting, one major challenge is that the Q-values of OOD actions are easily and incorrectly overestimated.
In this case, instead of using Q-values, we choose to trust the immediate rewards of each transition from environments, which are at least unbiased.
But only the immediate rewards maybe not be enough since our goal is to achieve higher accumulated returns and the greedy algorithm usually leads to a suboptimal policy.
On the other hand, only the Monte Carlo returns may also be not enough, because we do not know the coverage of the dataset (i.e., an action contained in a bad trajectory in given datasets still has the potential to achieve higher performance within other unseen trajectories).
Since ACQL should be able to apply to different kinds of tasks with different magnitudes of reward, it is better to use a normalized value ranging in (0, 1) to measure the data quality.
Combining the above ideas, we chose to combine both quantities and use $m(s,a) = \frac{1}{2} (r_{norm}(s, a) + g_{norm}(s, a))$ in Equation 9 to measure the qualities of in-dataset transition.

2.2 **Explanation for Equation 9**

In Equation 9, we show how we can measure the qualities of in-dataset actions.
Now the same question comes for OOD actions.
For a given state $s$ in dataset, an intuitive and common way is to use the $L_2$ distance to approximate the uncertainties.
A larger distance means that this OOD action generates a higher risk, i.e., we cannot trust it Q value too much.
Therefore we should suppress its Q value more.
Besides the $L_2$ distance, we should also consider the quality of the corresponding in-dataset action.
If the in-dataset action is actually a "good" one, even if the $L_2$ distance is higher, we can still trust the OOD action to some extent.
Combining the above ideas, now we can have:

$m(s, a_{\mu}) \propto m(s, a_{in}) - || a_{\mu} - a_{in} ||_{2}$

The rest terms and operations in Equation 9 are used to make the range in (0,1) and we use the mean value of $L_2$ distance along the action dimension.
Note that the action space is $a \in (-1, 1)$.

$\ \ \ \ \ || a_{\mu} - a_{in} ||_{2} \in (0, 2)$

$\Rightarrow \frac{1}{2} || a_{\mu} - a_{in} ||_{2} \in (0,1)$

$\Rightarrow m(s, a_{in}) - \frac{1}{2} || a_{\mu} - a_{in} ||_{2} \in (-1,1)$

$\Rightarrow m(s, a_{in}) - \frac{1}{2} || a_{\mu} - a_{in} ||_{2} + 1 \in (0,2)$

$\Rightarrow m(s, a_{\mu}) = \frac{1}{2} \left( m(s, a_{in}) -  \frac{1}{2} || a_{\mu} - a_{in} ||_{2} + 1 \right) \in (0,1)$

Now we can see the Equation 9 just looks complicated and its basic idea is intuitive and simple.

---

> ### Author Response · Authors · 2022-12-03
> **Response to Reviewers AX8y, 3P64, KtHP Cont'd**
>
>
> 2.3 **Explanation for Equations 10, 11**
>
> According to Equations 4 and 5, we know that how we can strictly control the ACQL Q-values compared to true Q-values and CQL over state-action point-wise.
> However, is it necessary to strictly control the AQL-Q values to the middle of true Q-values and CQL for every state-action pair?
> Actually, the answer is no.
> A simple example is that when $d_{\pi_{\beta}} > 0$, the Q-values of in-dataset actions will be larger than true Q-values as shown in the second term in Equation 3.
> Therefore we try to relax the conditions adaptively and use two variables $C_1, C_2$ to replace 0 in Equations 4 and 5 and form the losses $L_{cl-true}$ and $L_{cl-cql}$ in Equations 10 and 11.
>
> 2.4 **Explanation for Equations 12, 13**
>
> Now the question is how to calculate the variables $C_1, C_2$ adaptively.
> The basic idea here is that for a "good" action with higher $m(s,a)$, we should relax the conditions more, which means a smaller $C_1$ in Equation 4 (i.e., not much lower than true Q-values) and larger $C_2$ (i.e., much higher than CQL Q-values) in Equation 5.
> According to the above idea, now we can have:
>
> $C_1(s,a) \propto -m(s, a)$
>
> $C_2(s,a) \propto m(s, a)$
>
> Recall that $C_1, C_2$ are the Q-value gaps compared to true Q-values and CQL Q-values.
> Since for every different environment and dataset, the overall magnitude of Q-values is not the same.
> This is where the benefits of using the normalized quality measurements $m(s,a)$ come into play, we can easily scale and shift $m(s,a)$ to change its range more suitable for the current dataset.
>
> The last question is what is a suitable range of Q-values gap for each dataset.
> Firstly we know the range should be related to the Q-values.
> One simple choice is the upper bound of Q-values (i.e., $\frac{r_{max}}{1-\gamma}$) and then $C_1, C_2$ are calculated by
>
> $C_1 (s, a) = (1 - m(s, a)) \cdot Q_{max}$
>
> $C_2 (s, a) = m(s, a) \cdot Q_{max}$,
>
> where $Q_{max} = \frac{r_{max}}{1-\gamma}$.
> We evaluated this choice in practice and found it cannot work well as shown as ACQL($C_{Q-max}$) in the following Table.
> We argue that $Q_{max}$ is too large for the Q-value gap.
> Imagine that the difference between two Q-values is the upper bound of all Q-values, so it would relax the conditions too much in Equations 4 and 5, and lead to the invalidation of the restriction.
>
> How about the rewards of a single step?
> Following this idea, we tried two choices $r_{mean}$ and $r_{max}$, whose results are shown as ACQL($C_{r-mean}$) and ACQL in the following Table.
> Both rules work very well and lastly, we chose the better one using $r_{max}$ as the final version of ACQL implementation.
>
> 2.5 **Additional experiments for different rules.**
>
> To further evaluate the ACQL implementations with different rules, we conducted an extensive experiment and all results are in the following Table.
> We are also glad to add more experiments if you have more rules in mind and can describe them in more detail.
>
> Different rules for Equation 9.
>
> - ACQL(reward): We only use $m(s,a) = r_{norm}(s, a)$ to measure the qualities of data transitions.
>
> - ACQL(traj): We only use $m(s,a) = g_{norm}(s, a)$ to measure the qualities of data transitions.
>
> Different rules for Equations 12 and 13.
>
> - ACQL($C_{Q-max}$): As we mentioned above, we calculate $C_1, C_2$ by
>
> $C_1 (s, a) = (1 - m(s, a)) \cdot Q_{max}$
>
> $C_2 (s, a) = m(s, a) \cdot Q_{max}$,
>
> - ACQL($C_{r-mean}$): As we mentioned above, we calculate $C_1, C_2$ by
>
> $C_1 (s, a) = (1 - m(s, a)) \cdot r_{mean}$
>
> $C_2 (s, a) = m(s, a) \cdot r_{mean}$,
>
> - ACQL($C_0$): We also tried to directly set $C_1, C_2=0$, which means that the restrictions in Equations 4 and 5 should be satisfied strictly.
>
> From the results shown in the following Table, we can easily observe that all rules except ACQL($C_{Q-max}$) can work very well on all tasks.
> As we mentioned before, $Q_{max}$ in ACQL($C_{Q-max}$) is too large for the Q-value gap and leads to invalidation of the restriction in Equations 4 and 5.

---

> > ### Author Response · Authors · 2022-12-03
> > **Response to Reviewers AX8y, 3P64, KtHP Cont'd**
> >
> > Table:  Comparision among differnt rules of ACQL on D4RL Gym-MuJoCo v2 environments.
> > |  Task Name  | CQL[1,2] | ACQL(reward) | ACQL(traj) | ACQL($C_{Q-max}$) | ACQL($C_{r-mean}$) | ACQL($C_0$) | ACQL |
> > |  :----:  | :----:  | :----:  | :----:  | :----:  | :----:  | :----: | :----: |
> > | halfcheetah-m-e-v2 | 91.6 | 86.3 | **94.4** | 26.2 | 90.2 | 93.9 | 87.9 |
> > | halfcheetah-m-r-v2 | 45.5 | 46.3 | 44.9 | 21.7 | 47.7 | **52.3** | 46.3 |
> > | halfcheetah-m-v2 | 44.0 | 47.2 | **52.8** | 20.1 | 47.5 | 50.3 | 48.4 |
> > | halfcheetah-r-v2 | 18.6 | 23.0 | **27.5** | 2.3 | 9.2 | 10.1 | 25.9 |
> > | halfcheetah-sum | 199.7 | 202.8 | **219.6** | 70.3 | 194.6 | 206.6 | 208.5 |
> > |  ----  | ----  | ----  | ----  | ----  | ----  | ----  | ----  | ---- |
> > | hopper-m-e-v2 | **105.4** | 84.3 | 92.2 | 13.4 | 95.6 | 83.6 | 82.2 |
> > | hopper-m-r-v2 | 95.0 | 95.5 | **97.9** | 40.9 | 96.6 | 95.4 | **97.9** |
> > | hopper-m-v2 | 58.5 | 80.8 | 84.8 | 28.3 | 82.9 | 77.4 | **91.4** |
> > | hopper-r-v2 | 9.3 | 7.4 | 5.6 | 6.2 | 7.3 | 8.8 | **31.4** |
> > | hopper-sum | 268.2 | 268.0 | 280.5 | 88.8 | 282.4 | 265.2 | **302.9** |
> > |  ----  | ----  | ----  | ----  | ----  | ----  | ----  | ----  | ---- |
> > | walker-m-e-v2 | 108.8 | 111.2 | **115.3** | 16.3 | 112.4 | 111.5 | 108.3 |
> > | walker-m-r-v2 | 77.2 | 62.6 | 74.5 | 30.3 | **86.1** | 82.2 | 85.5 |
> > | walker-m-v2 | 72.5 | 78.5 | 83.6 | 18.1 | 84.6 | **87.3** | 84.9 |
> > | walker-r-v2 | 2.5 | **6.9** | 4.8 | 8.0 | 0.0 | 1.6 | 5.2 |
> > | walker-sum | 261.0 | 259.2 | 278.2 | 72.7 | 283.1 | 282.6 | **283.9** |
> >
> > [1] Prudencio, Rafael Figueiredo, Marcos ROA Maximo, and Esther Luna Colombini. "A Survey on Offline Reinforcement Learning: Taxonomy, Review, and Open Problems." arXiv preprint arXiv:2203.01387 (2022).
> >
> > [2] Kostrikov, Ilya, et al. "Offline reinforcement learning with fisher divergence critic regularization." International Conference on Machine Learning. PMLR, 2021.

---

### Author Response · Authors · 2022-12-07
**Response to Reviewers AX8y, 3P64**

We really appreciate your positive and constructive comments and suggestions, which have greatly helped us improve the paper.
After we added so many additional experiments and detailed explanations about the design rules of ACQL implementations, we think a major concern is that our ACQL failed on AntMaze tasks.
Please allow us to give an explanation of what we think about these results.

**Regarding the results of AntMaze tasks.**

- One major reason for the failures is that the tasks on the AntMaze environments are sparse reward problems and the rewards are only 0 or 1.
Thus it is too difficult to use Equations 8 and 9 to calculate accurate quality measurements for each transition, which leads to task failures.
However, following the current design philosophy of ACQL implementation, which tries to get the adaptive conservative weights using relatively quality measurements, these results do not surprise us and it is indeed an inherent flaw in the current implementation.

- As mentioned in the Introduction Section, our main goal is to design a balanced algorithm for datasets with different data qualities (e.g., expert, medium, and random datasets), so it is out of scope to require current ACQL implementation additionally to solve the sparse reward problem, which is another important and challenging field in DRL research.

- We argue that the AntMaze tasks are very special since most of the other baselines including Behavior Cloning, Decision Transformer, AWAC, OnestepRL, TD3+BC also failed in these tasks (e.g., achieved around 0 success rates on antmaze-large datasets) as shown in Table 9 in Appendix C.3.
We believe that these algorithms are in a similar situation to ours in that none of them are also considered separately for the sparse reward problem, making them hard to work on the AntMaze tasks.
However, we think we cannot deny their contribution just because they failed in particular scenarios and it is the same for our paper.
In addition, the current ACQL implementation can achieve better and comparable performance on Gym-MuJoCo, Adroit, and Franka Kitchen tasks (i.e., around 30 tasks in total), which has demonstrated its effectiveness.

- Note that CQL is also a special instance of ACQL framework (i.e., by setting the adaptive weights directly using a fixed $\alpha$), its success on AntMaze tasks shows that the compatibility and flexibility of ACQL framework, under which every user can design their own adaptive weight functions for different kinds of scenarios.
As mentioned in the last paragraph in Section 5.1, we never claim the current implementation is the only and best solution and we would welcome any continual improvement under the umbrella of ACQL.
For example, if we have already known the task is a sparse reward problem, we can design a special adaptive weight function in a similar way to the CQL.

- We also want to emphasize the main contribution in our paper is not only the current ACQL implementation, but also the whole ACQL framework in Section 4.
The current ACQL framework provides a very flexible tool and corresponding theoretical analysis to sheds light on how to design a proper conservative Q-function.
As mentioned in Section 4 and Appendix A, the ACQL framework supports controlling the conservative levels in many kinds of granularity (e.g., control over each state-action pair, each state, a part of transitions, or even the whole empirical MDP).
So it essentially converts the problem in offline DRL into how to find the most appropriate conservative levels for the critic function.
This direction is yet to be fully explored and thus has huge potential for improvement in offline DRL, while the ACQL framework has already taken one step towards it.
Instead of only providing an effective method, we sincerely hope it can inspire you to some extent, which is a very important factor in judging a paper.


In summary, we argue that it is **unreasonable**, **unfair**, and **too demanding** to require our current ACQL implementation to outperform other baselines on such sparse reward tasks.
Even though the current implementation has some inherent flaws in particular scenarios, this does not affect the other contributions of our paper, including the ACQL framework and our strong results over around 30 tasks on Gym-MuJoCo, Adroit, and Franka Kitchen domains.

---

### Decision · Program_Chairs · 2023-01-20

**Decision:**

Reject

**Justification For Why Not Higher Score:**

There are weaknesses in the empirical performance of the proposed method (e.g. only small improvements over CQL, instability, poor performance on antmaze) and the theoretical analysis. None of these weaknesses are a dealbreaker on their own; e.g. I agree that gaps between theory and practice are expected, but the weaknesses seem to outweigh the strengths in the current version.

**Justification For Why Not Lower Score:**

N/A

**Metareview: Summary, Strengths And Weaknesses:**

There are definitely some interesting ideas in this paper, but that there is not enough reviewer support to warrant acceptance. In particular, there are weaknesses in the empirical performance of the proposed method (e.g. only small improvements over CQL, instability, poor performance on antmaze) and the theoretical analysis. None of these weaknesses are a dealbreaker on their own; e.g. I agree that gaps between theory and practice are expected, but the weaknesses seem to outweigh the strengths in the current version.